



# Importance of aerosols and shape of the cloud droplet size distribution for convective clouds and precipitation

Christian Barthlott[1], Amirmahdi Zarboo[1], Takumi Matsunobu[2], and Christian Keil[2]

[1]Institute of Meteorology and Climate Research (IMK-TRO), Department Troposphere Research, Karlsruhe Institute of Technology (KIT), Karlsruhe, Germany
[2]Meteorologisches Institut, Ludwig-Maximilians-Universität, Munich, Germany

**Correspondence:** Christian Barthlott (christian.barthlott@kit.edu)

**Abstract.** The predictability of deep moist convection is subject to large uncertainties resulting from inaccurate initial and boundary data, the incomplete description of physical processes, or microphysical uncertainties. In this study, we investigate the response of convective clouds and precipitation over central Europe to varying cloud condensation nuclei (CCN) concentrations and different shape parameters of the cloud droplet size distribution (CDSD), both of which are not well constrained by

observations. We systematically evaluate the relative impact of these uncertainties in realistic convection-resolving simulations for multiple cases with different synoptic controls using the new icosahedral nonhydrostatic ICON model. The results show a large systematic increase in total cloud water content with increasing CCN concentrations and narrower CDSDs together with a reduction in the total rain water content. This is related to a suppressed warm-rain formation due to a less efficient collision-coalescence process. It is shown that the evaporation at lower levels is responsible for diminishing these impacts on surface

precipitation, which lies between +13% to -16% compared to a reference run with continental aerosol assumption. In general, the precipitation response was larger for weakly-forced cases. We also find that the overall timing of convection is not sensitive to the microphysical uncertainties applied, indicating that different rain intensities are responsible for changing precipitation totals at the ground. Furthermore, weaker rain intensities in the developing phase of convective clouds can allow for a higher convective instability at later times, which can lead to a turning point with larger rain intensities later on. The existence of such

a turning point and its location in time can have a major impact on precipitation totals. In general, we find that an increase in the shape parameter can produce almost as large a variation in precipitation as a CCN increase from maritime to polluted conditions. Narrowing of the CDSD not only decreases the absolute values of autoconversion and accretion, but also decreases the relative role of the warm-rain formation in general, independent of the prevailing weather regime.

We further find that increasing CCN concentrations reduces the effective radius of cloud droplets stronger than larger shape

parameters. The cloud optical depth, however, reveals a similar large increase with larger shape parameters as changing the aerosol load from maritime to polluted. By the frequency of updrafts as a function of height, we show a negative aerosol effect on updraft strength, indicating that the larger water load above the freezing level in polluted conditions does not lead to an invigoration of deep convection. These findings demonstrate that both, CCN assumptions and the CDSD shape parameter, are important for quantitative precipitation forecasting and should be carefully chosen if double-moment schemes are used for

modeling aerosol-cloud interactions.



# 1 Introduction

Despite recent improvements in numerical weather forecasting by, e.g., higher grid spacing, improved parameterizations of physical processes, ensemble modeling strategies or post-processing techniques, the accurate forecast of convective precipitation is still a challenge for state-of-the-art numerical weather prediction (NWP) models. Cloud formation and subsequent

precipitation results from a chain of complex processes in the atmosphere and is therefore accompanied by numerous uncertainties in its formation (e.g. Schneider et al., 2019). Many aspects influence the predictability of convective precipitation, e.g. the synoptic-scale flow, the presence of mountains, and the heterogeneity of the land surface. In current convection-permitting ensemble modeling systems, the uncertainties in the initial and lateral boundary conditions as well as uncertainties in the representation of physical processes are accounted for (e.g., Clark et al., 2016; Barthlott and Barrett, 2020, and references therein).

Large uncertainties also arise from the nonlinear character of the microphysics and the complexity of the microphysical system with many possible process pathways (Seifert et al., 2012; Schneider et al., 2019).

Aerosol-cloud interactions are considered to be one of the most uncertain processes in NWP models (e.g. Tao et al., 2012; Altaratz et al., 2014; Fan et al., 2016; Barthlott and Hoose, 2018). In general, it is assumed that the activation of aerosol particles, acting as cloud condensation nuclei (CCN), into cloud droplets leads to more numerous and smaller cloud droplets

in situations with high aerosol load. The smaller droplet size then suppresses the onset of precipitation in warm clouds due to a reduction of the collision-coalescence process, leading to a longer cloud lifetime ("lifetime effect", Albrecht, 1989). In polluted conditions, the larger water load at the freezing level can result in an additional release of latent heat which can lead to an invigoration of the convective clouds with increased precipitation amounts (Rosenfeld et al., 2008). The impact of aerosols on cloud formation and precipitation has been shown to differ between cloud types, the aerosol regime, and environmental

conditions (e. g. Seifert and Beheng, 2006b; Khain et al., 2008; van den Heever et al., 2011; Tao et al., 2012; Barthlott et al., 2017). In a model intercomparison effort for a convective case near Houston (Texas), Marinescu et al. (2021) demonstrated that the participating models showed several consistent trends, but the change in the amount of deep convective updrafts through varying CCN concentrations varies significantly. These differences may be related to the differences in the evolution of the environmental conditions within the models. By comparing the relative contributions of varied aerosol concentrations, soil

moisture heterogeneities, and a stochastic boundary-layer perturbation scheme for a 10-day period of high-impact weather in Central Europe, Keil et al. (2019) found that perturbed aerosol concentrations impact the spatial precipitation variability already from the model start onwards, but to a smaller degree than the other perturbations. In the study by Barthlott and Hoose (2018), a novel technique to modify the environmental atmospheric conditions in realistic simulations was introduced. They modified the initial and boundary temperature profiles with a linearly increasing increment for six cases classified into weak and strong

synoptic-scale forcing. Results show that more accurate environmental conditions are more important than accurate aerosol assumptions, especially for weak forcing. The aerosol effect, however, is non-negligible, systematic for strongly forced cases, but non-systematic and largest for weakly forced cases. The decrease of total precipitation with increasing aerosol load for strong synoptic forcing was due to the suppression of the warm-rain process, also documented by, e.g., Tao et al. (2012); Storer and van den Heever (2013). By means of idealized simulations, Grant and van den Heever (2015) showed that the altitude of





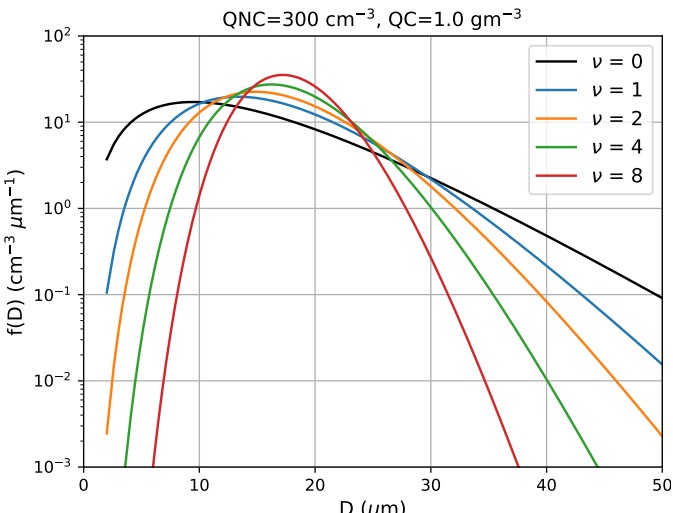

**Figure 1.** Cloud droplet size distributions for different values of the shape parameter $\nu$ at fixed cloud water content (QC) and cloud droplet number concentration (QNC). $D$ denotes the diameter of the droplets.

dry layers is important for the development of deep convective clouds and that the impact of aerosols varies inversely with the storm organization. Barthlott et al. (2017) similarly highlighted the importance of evaporation of rain drops in simulations for the 2014 Pentecost storm over Germany. They found a systematic relationship for condensate amounts of cloud water, rain and ice with increasing CCN, but evaporation at lower levels lead to a non-systematic response of accumulated precipitation.

Another source of uncertainty lies in the width of the cloud droplet size distribution (CDSD), as CCN conditions also
influence the size distribution of the nucleated droplets. The underlying generalized Gamma distribution

$$f(x) = Ax^{\nu}\exp(-\lambda x^{\mu}) \tag{1}$$

depends on the shape parameter $\nu$ and dispersion parameter $\mu$ as a function of the particle mass $x$. $A$ and $\lambda$ can be calculated from the predicted mass and number densities (Seifert and Beheng, 2006a). The shape parameter controls the width of the size distribution which has important implications on microphysical processes, e.g., autoconversion and evaporation. A higher
shape parameter is supposed to suppress autoconversion (e.g. Seifert and Beheng, 2001), leading to higher droplet number concentrations. The droplet size distribution also plays a crucial role in determining the radiative properties of clouds. The width of the CDSD is not well constrained by observations, but important for accurately predicting condensation and evaporation rates (Igel and van den Heever, 2017a). Previous observational studies showed the large range in shape parameters values based on cloud type and environmental conditions ranging between 0–14 (e.g. Levin, 1958; Gossard, 1994; Miles et al., 2000;
Martins and Silva Dias, 2009). Figure 1 presents Gamma size distributions with different shape parameters for a fixed cloud water content (QC) and cloud droplet number concentration (QNC) and shows that increasing the shape parameter narrows the size distribution. An important point is the fact that with a low shape parameter, the curves show more small droplets, but also more large droplets leading to a larger effective radius, which is the relevant parameter for determining the cloud optical





properties. This was also documented by Morrison and Grabowski (2007), who found that higher shape parameters result in
a decrease of the effective radius. Previous modeling studies on the impact of the shape parameter are rare and were mostly
based on idealized simulations. For example, using large-eddy simulations of non-precipitating shallow cumulus clouds, Igel
and van den Heever (2017c) have shown that evaporation rates are much more sensitive to the values of the shape parameter
than to the condensation rates. As a result, cloud properties such as droplet number concentration, mean droplet diameter,
and cloud fraction were strongly impacted and changes were found to be on the same order of magnitude as changes due to
increasing or decreasing the aerosol concentration by a factor of 16 (Igel and van den Heever, 2017c). This documents (i) a need
to further assess the impacts of the chosen CDSD parameters on various cloud types in different weather regimes and (ii) to
evaluate the suitability of including CDSD uncertainties in ensemble forecasting. We therefore expand this line of investigation
by disturbing the shape parameter of the CDSD for precipitating clouds in real-case simulations. In addition, different aerosol
concentrations ranging from clean to polluted conditions are assessed. By comparing the effect of the ambient aerosol amount
to changes of the shape parameter, we can quantify their relative impact on the forecast of convective precipitation. To cover
different weather regimes, these analyses will be conducted for situations with both weak and strong synoptic forcing. Another
goal of this study is to determine if and how the large range of possible shape parameters from observations (see e.g. Tab. 1 in
Igel and van den Heever, 2017b) impacts the simulation results. Moreover, we want to investigate if the sensitivity to the shape
parameter can induce larger precipitation changes than the ones from different CCN concentrations. The unique aspect of this
work is the fact that it is the first to systematically evaluate the relative impact of CCN concentrations and uncertainties in the
CDSD for multiple cases with different synoptic controls.

## 2 Method

### 2.1 Model description and simulations overview

The numerical simulations of this study were conducted with version 2.6.2.2 of the ICOsahedral Non-hydrostatic (ICON)
model. ICON is based on an icosahedral-triangular Arakawa-C grid with grid-nesting capability which can be run in global
and limited-area mode (Zängl, 2012). The prognostic variables are the horizontal velocity component normal to the triangle
edges $v_\mathrm{n}$, the vertical wind component $w$, density $\rho$, and virtual potential temperature $\theta_\mathrm{v}$. Time integration is performed with a
two-time-level predictor-corrector scheme that is fully explicit in the horizontal and implicit for the terms describing vertical
sound wave propagation. For a detailed description of the non-hydrostatic dynamical core, we refer to Zängl et al. (2015). At
the German Weather Service (DWD), the ICON model is operational on the global scale since January 2015 (resolution 13 km)
and since July 2015, the two-way nested configuration ICON-EU for Europe with 7 km resolution is available. Since February
2021, the convection-permitting version ICON-D2 has replaced the former model COSMO-D2 (COnsortium for Small-scale
MOdeling) for conducting operational forecasts over Germany.

The unstructured triangular grid is based on successive refinement of a spherical icosahedron. In this study, we use a so-called
R19B07 grid which has 538164 cells on our simulation domain (Fig. 2). The effective horizontal grid spacing corresponds to
2 km. We use a height-based terrain-following coordinate system based on the smooth level vertical (SLEVE) coordinate





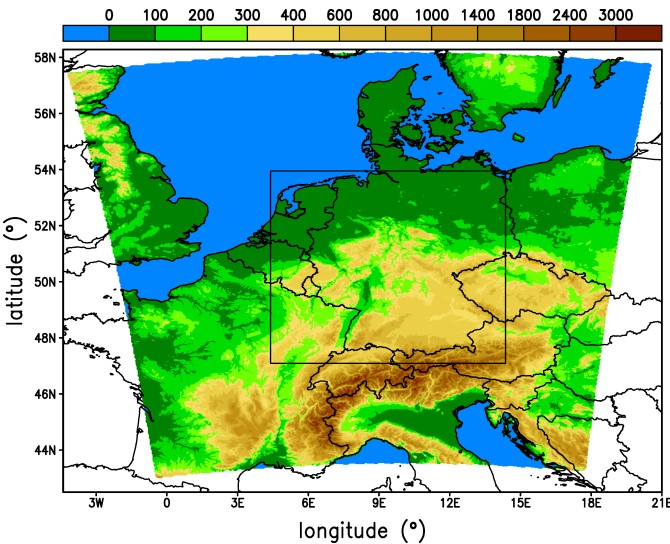

**Figure 2.** ICON simulation domain and model orography in meters above sea level. The black rectangle depicts the evaluation domain covering most of Germany and parts of neighboring countries.

implementation (Leuenberger et al., 2010) with 65 levels. Model domain, horizontal and vertical resolution correspond to the operational ICON-D2 configuration. For the simulation of aerosol effects on mixed-phase clouds, we use the two-moment microphysics scheme of Seifert and Beheng (2006a) which enables the use of four different CCN concentration assumptions.

This scheme predicts mass and number concentration of cloud water, rain water, ice, snow, graupel, and hail. As already documented in Barthlott and Hoose (2018), the activation of CCN from aerosol particles is computed using pre-calculated activation ratios stored in look-up tables by Segal and Khain (2006). For further details about the activation used here, we refer the interested reader to Barthlott et al. (2017) or Barthlott and Hoose (2018). To investigate aerosol-cloud interactions, we performed numerical simulations with maritime (number density $N_{CN}$ = 100 cm$^{-3}$), intermediate ($N_{CN}$ = 500 cm$^{-3}$),

continental ($N_{CN}$ = 1700 cm$^{-3}$), and continental polluted conditions ($N_{CN}$ = 3200 cm$^{-3}$). Hande et al. (2016) documented that the continental aerosol assumption represents typical conditions of central Europe. Instead of an explicit supersaturation prediction, the ICON model uses a saturation adjustment scheme to predict droplet condensation, similar to many other double-moment schemes (e. g. Cohard and Pinty, 2000; Milbrandt and Yau, 2005; Morrison et al., 2009; Dipankar et al., 2015). The application of a saturation adjustment technique is considered to be appropriate because almost all clouds (except extremely

maritime ones) relax rapidly to the thermodynamic equilibrium between water vapor and water drops (Seifert and Beheng, 2006a).

Further physics parametrizations include a multi-layer land-surface scheme Terra (Heise et al., 2006), a turbulence scheme based on a prognostic equation of the turbulent kinetic energy (Raschendorfer, 2001), and a rapid radiation transfer model (RRTM, Mlawer et al., 1997) for radiation. Applied with 2-km grid spacing, deep convection is resolved, but shallow con-

vection still needs to be parameterized using the Tiedtke-Bechtold shallow convection scheme (Bechtold et al., 2008; Tiedtke,





**Table 1.** Overview of the numerical simulations.

| Name | CCN concentration | Shape parameter |
|------|-------------------|-----------------|
| m0 | maritime | 0 |
| i0 | intermediate | 0 |
| c0 (=REF) | continental | 0 |
| p0 | continental polluted | 0 |
| c1 | continental | 1 |
| c2 | continental | 2 |
| c4 | continental | 4 |
| c8 | continental | 8 |

1989) which is able to generate small amounts of convective precipitation. In contrast to the Tiedtke-scheme used in COSMO, this scheme has been tuned to avoid excessive moisture transport out of the boundary layer, which reduces a dry bias in the boundary layer. Depending on the day of investigation (see section 2.2), we use either analyses from the Integrated Forecast System (IFS) of the European Centre for Medium-Range Weather Forecasts (ECMWF), or from ICON-EU as initial and

boundary data. All simulations were initialized at 00:00 UTC with an integration time of 24 h.

For each of the investigated cases (described in the next section), we conducted eight simulations (Tab. 1): a first set with four different CCN concentrations using a reference shape parameter of 0 and a second set with four different values of the shape parameter using the reference CCN concentration (i.e. the continental aerosol assumption). We apply the same range of shape parameter values as in idealized simulations of Wellmann et al. (2020).

## 2.2 Case studies

To cover different typical weather regimes in central Europe, we consider cases with convective precipitation under weak and strong synoptic-scale forcing. We performed numerical simulations for six days in total, three for each synoptic-scale forcing class (Tab. 2). The two cases of 2016 belong to an exceptional sequence of severe thunderstorms in Germany. The meteorological background of this high-impact weather period is described by Piper et al. (2016). In a recent publication by

Keil et al. (2019), different aspects of predictability (i.e. soil moisture heterogeneities, a stochastic boundary-layer perturbation scheme and varied aerosol concentrations) for this period were analyzed. The cases of 1 July 2009 and 11 September 2013 were also simulated with the COSMO model (500-m grid length) to study the relative impact of soil moisture, CCN concentrations and terrain forcing by Schneider et al. (2018, 2019). The weakly forced case of 9 June 2018 belongs to a long-lasting episode with severe thunderstorms described in more detail by Mohr et al. (2020). Wilhelm et al. (2021) gives a detailed description

of the synoptic controls of the 3-day storm series in June 2019 to which our last case (11 June) belongs to. To evaluate the synoptic-scale forcing quantitatively, we calculated the convective adjustment time scale following the approach of Keil et al. (2014). If the daily mean of this time scale is larger than a threshold of 3 h, the synoptic-scale forcing is weak, lower values



**Table 2.** List of case studies and convective adjustment time scale $\tau$.

| Large-scale forcing | Date | $\tau$ (h) |
|---|---|---|
| weak | 01 July 2009 | 6.26 |
| weak | 05 June 2016 | 5.22 |
| weak | 09 June 2018 | 4.65 |
| strong | 11 September 2013 | 0.13 |
| strong | 02 June 2016 | 1.45 |
| strong | 11 June 2019 | 2.00 |

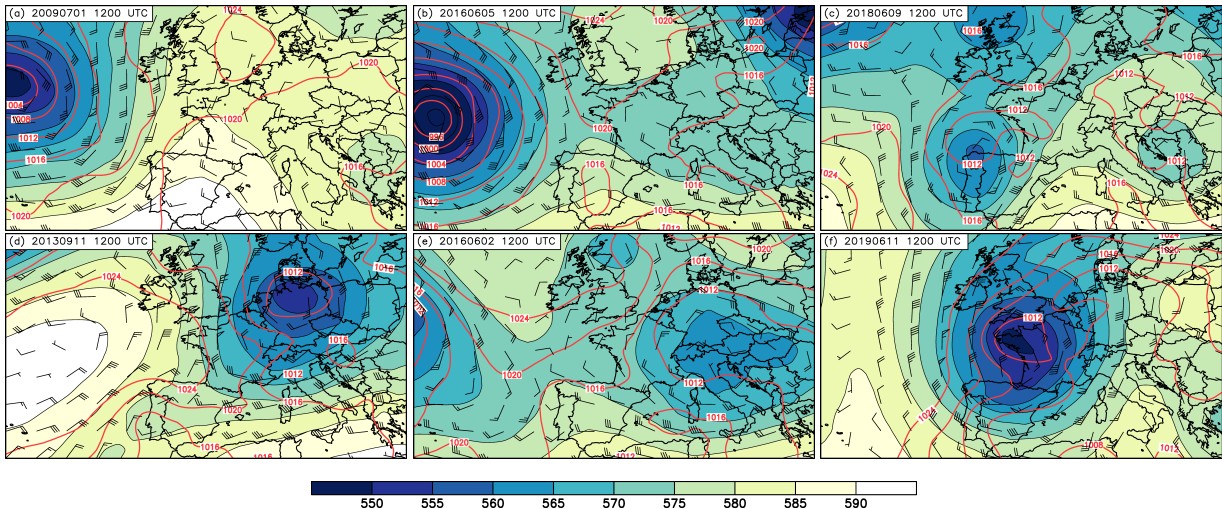

**Figure 3.** GFS analyses at 1200 UTC for the cases with weak (top) and strong (bottom) synoptic-scale forcing showing 500-hPa geopotential height (gpdm; shading), sea-level pressure (hPa, red contours), and 500-hPa wind barbs.

indicate strong forcing. The resulting values (Tab. 2) agree well with the results from visual inspection of synoptic weather charts (Fig. 3).

All cases with weak synoptic forcing show a dominating ridge in central Europe (top panels in Fig. 3). The axis of the ridge lies over France for the cases from 2009 and 2016 and further to the east on 9 June 2018. Over the eastern Atlantic, low pressure systems are present and over Germany, mid-tropospheric winds are weak with northerly (a), easterly (b), and southwesterly (c) winds. The surface pressure ranges between 1012–1020 hPa with weak horizontal gradients. The total precipitation amount of the respective reference runs (i.e. with continental CCN concentrations and a cloud droplet shape parameter of 0) shows scattered convection over Germany for these cases (Fig. 4a–c).

The flow of the cases with strong synoptic forcing shows a stronger baroclinicity (bottom panels of Fig. 3). A low-pressure system is situated over Germany on 11 September 2013 and on 2 June 2016. The mid-tropospheric flow is cyclonic with

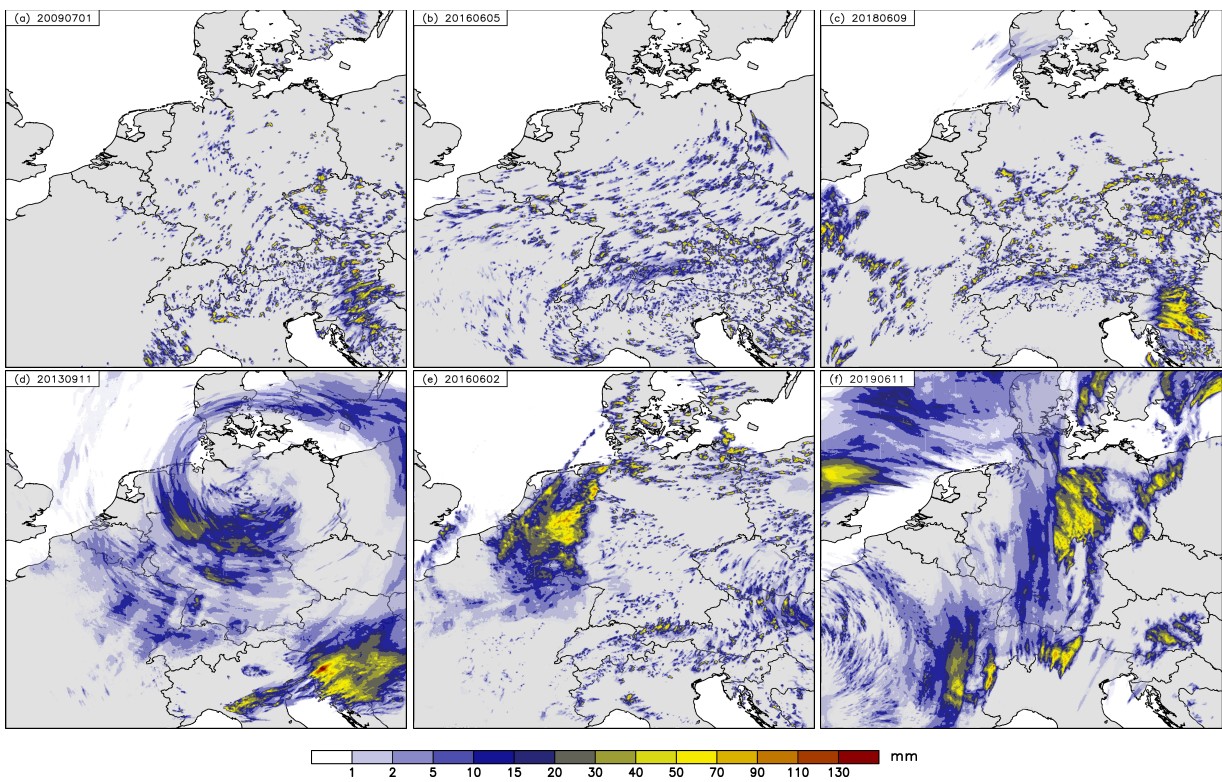

**Figure 4.** 24-h precipitation amount of the reference runs with continental CCN concentration and broad cloud droplet size distribution ($\nu = 0$) for the cases with weak (top) and strong (bottom) synoptic-scale forcing.

stronger winds on the first case due to a deeper low and stronger pressure gradients. The case of 11 June 2019 is characterized by a trough over western France. On that day, Germany lies downstream of this trough in a strong southerly flow. The cyclonic

circulation on 11 September 2013 is also visible in the precipitation pattern of that day (Fig. 4d). Within this frontal precipitation, convective showers are embedded. For the remaining two cases with strong synoptic forcing (2 June 2016 and 11 June 2019), larger cloud clusters are simulated over Germany as well (Figs. 4e–f).

## 3 Results

### 3.1 Precipitation amount and timing

At first, we analyze the 24-h accumulated precipitation amount which was computed for the Germany domain depicted by the black box in Fig. 2. The results show that the cases with strong synoptic forcing all have higher precipitation totals than the ones with weak synoptic-scale forcing (Fig. 5). The precipitation deviations from the respective reference runs (given in the lower panels) show larger variations for the weakly-forced cases (+13% to -16%) than in the strongly-forced cases (+7% to



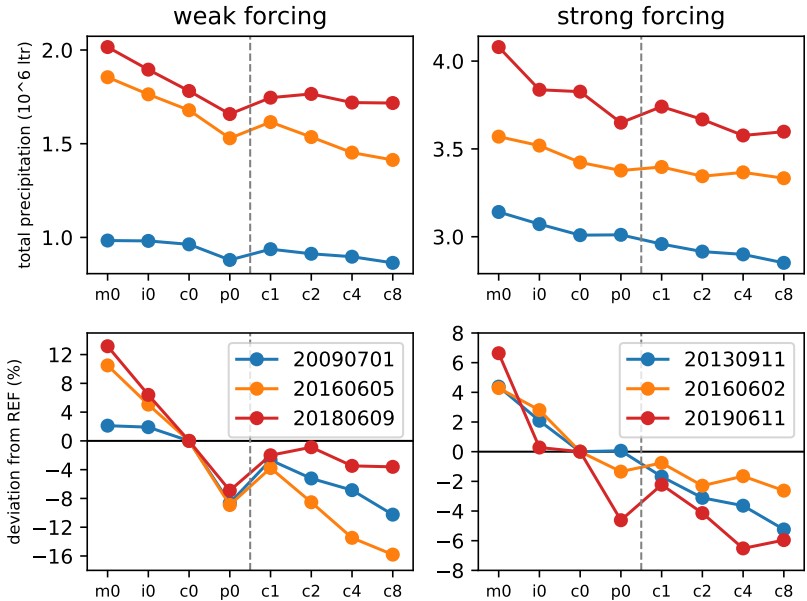

**Figure 5.** Domain-accumulated precipitation (top) and precipitation deviation from the respective reference run (bottom) for cases with weak (left) and strong (right) synoptic-scale forcing. Runs with increasing CCN concentrations are labeled m0–p0, whereas runs with increasing shape parameter and continental aerosol assumptions are labeled c1–c8. The reference run is depicted as c0.

-7%). This is in agreement with previous findings investigating convective precipitation in central Europe using the COSMO
model (Barthlott and Hoose, 2018; Schneider et al., 2019; Keil et al., 2019).

An important finding is the fact that only maritime and intermediate CCN concentrations lead to a precipitation enhancement and that for all days irrespective of the synoptic forcing, the precipitation amount decreases with increasing CCN concentration. This points towards the influence of a reduced warm-rain process which will be analyzed later by the analysis of microphysical process rates. The runs with higher shape parameters all show less precipitation than the reference run and almost all days
show a systematic precipitation decrease with increasing shape parameter. It is further of interest to see if changes in the shape parameter (runs c1, c2, c4, c8) have a larger impact on the precipitation amounts than the CCN concentration (runs m0, i0, c0, p0). For the cases analyzed in this study, we see a larger absolute deviation (i.e. the range between maximum and minimum precipitation deviation) of 20.1% for varying CCN concentrations, whereas changes in the shape parameter yield to a maximum difference of 15.8%. Also, the mean changes of all days is higher for CCN variations (11.9%) than for shape parameter changes
(7.3%). There is only one case (11 September 2013) where changing the shape parameter leads to larger total deviations (5.2%) than do the different CCN concentrations (4.3%). However, this strong forcing day has somehow weaker convective activity with more stratiform precipitation which could explain this differing behavior. In contrast to the CCN response, changes in the shape of the CDSD always leads to a precipitation reduction with respect to the reference run.

It is natural to ask if the precipitation response to CCN in ICON is similar as in previous COSMO simulations using the same
two-moment microphysics scheme. Findings from Barthlott and Hoose (2018) revealed a systematic precipitation decrease with





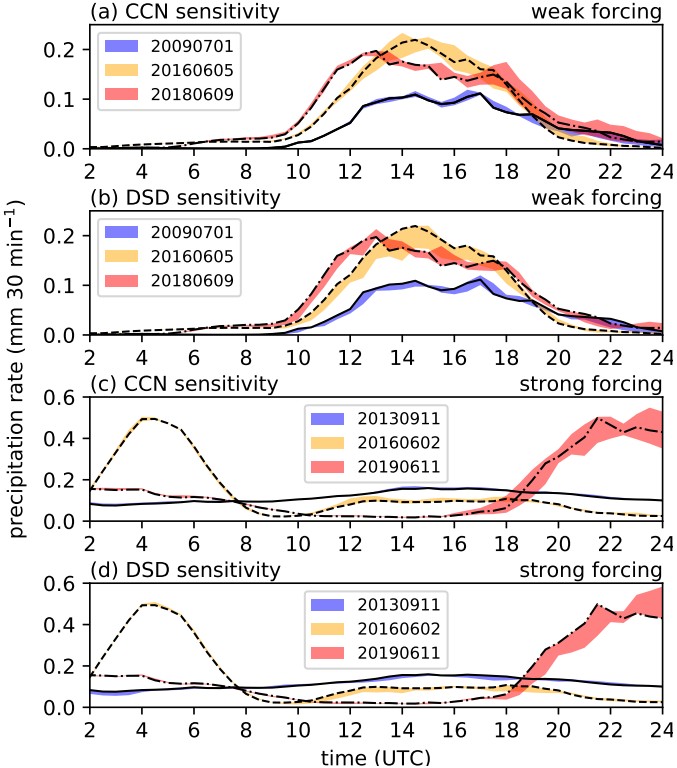

**Figure 6.** Domain-averaged precipitation rate for weak forcing (a, b) and strong forcing (c, d). The CCN sensitivities are shown in (a) and (c), the CDSD sensitivities in (b) and (d). The black lines indicate the respective reference run with continental CCN concentration and a shape parameter of 0.

increasing CCN concentration only for cases with strong synoptic forcing, whereas no systematic relationship was found for weakly forced conditions. Barthlott et al. (2017) and Schneider et al. (2019) also found a non-systematic precipitation response. The systemic precipitation decrease found in this study for all days is therefore remarkable. Whether this difference is case-dependant or caused by a different implementation of the scheme in the ICON model (e.g. saturation adjustment before and after the microphysics) cannot be answered here and is left for future work.

We have also computed domain-averaged half-hourly precipitation rates and found that there is little significant difference among the sensitivity runs with respect to the timing of convective precipitation (Fig. 6). All different model configurations have similar precipitation onset and decay times. Although individual clouds may have started to precipitate earlier or later, at least on average over the evaluation domain of Germany, the timing is similar. There are, however, differences in the magnitude

of precipitation intensities. As the different CCN assumptions and shape parameter values do not have an impact on the mean timing of precipitation, the different precipitation totals are solely generated by varying rain intensities in our evaluation domain. As was already obvious from the precipitation amounts presented in Fig. 5, the days with weak forcing show, on average, a larger spread in precipitation intensities. The strong forcing case of 11 June 2019, however, shows the largest spread





of all cases analyzed, but only after 18:00 UTC. The general trend of decreased precipitation totals with narrower CDSD is
also obvious from the respective reference runs, which mostly lie at the upper end of shaded range of rain intensities in panels
(b) and (d) of Fig. 6.

To better quantify differences in the rain intensity over the course of the 24-h simulation period, we computed the fraction
of 30-min intervals having a higher and lower rain rate as the reference run with continental CCN concentration and shape
parameter of 0 (not shown). Averaged over all 6 cases analyzed, there is a clear dominance of stronger rain intensities for
maritime CCN (run m0, 85% of time) and intermediate CCN (run i0, 79% of time). The CDSD sensitivity runs have larger
precipitation rates in only 26–33% of the time which explains the reduced precipitation totals after 24 h.

Within this context, the case of 11 June 2019 shows another interesting feature. While precipitation rates of the model runs
with increased shape parameters for that day are all lower than the reference run between 1800–2200 UTC, they show larger
rain intensities after that until midnight (Fig. 6d). This can be explained by the fact that the weaker convection before 2200 UTC
is not consuming as much convective available potential energy (CAPE) as the runs with higher rain intensities. The temporal
evolution (not shown) reveals that CAPE in the CCN-runs decreases stronger with time after 1800 UTC than the CDSD-runs.
Similarly, the lapse rates between 700–500 hPa are slightly steeper in that period (not shown). As convection decays in the
CCN-runs, the higher instability and potential energy in the CDSD-runs then leads to a turning point where those runs have
higher rain rates and convection is still in its mature stage. The short available time until the end of the simulation period
hinders these runs from getting similar or even higher total rain amounts than the CCN runs at the end. This interaction of the
convective development and the available instability can be an important point in other cases as well as the total precipitation
amounts can depend on whether (i) such a turning point exists and (ii) if it occurs early enough to have a major impact on
precipitation totals. This could be considered as another "lifetime effect", similar to the one proposed by Albrecht (1989) about
the suppressed onset of precipitation in warm clouds due to a weaker collision and coalescence of cloud droplets forming rain
drops in polluted conditions. Here, however, this is more related to the thermodynamics and prolonged instability.

## 3.2 Convection-related parameters and cloud fraction

As environmental conditions such as instability, CAPE, or relative humidity were shown to be more important for precipitation
totals than accurate aerosol assumptions (Barthlott and Hoose, 2018), it is important to assess their evolution in our model runs.
Therefore, we have computed the temporal evolution of several convection-related parameters for all model runs (not shown). It
turns out that the environmental conditions in which the first clouds and subsequent precipitation form are very similar, at least
for domain averages over Germany. Thus, the sensitivity runs comprising different CCN concentrations and different shapes
of the CDSD, do not modify the initial environmental conditions. However, the microphysical uncertainties do impact the
cloud structure and precipitation rates, which themselves influence the environment in the vicinity of the clouds. For example,
CAPE and convective inhibition are very similar until the mature stage of convection, differences occur only later as a result of
different precipitation intensities. Another example is the convergence of the low-level wind, which is an important parameter
describing the impact of weaker/stronger cold pools on the initiation or intensification of secondary cells (Barthlott and Hoose,
2018). For this parameter as well, we do not see any distinct differences in the initiation phase and evolving precipitation. Only





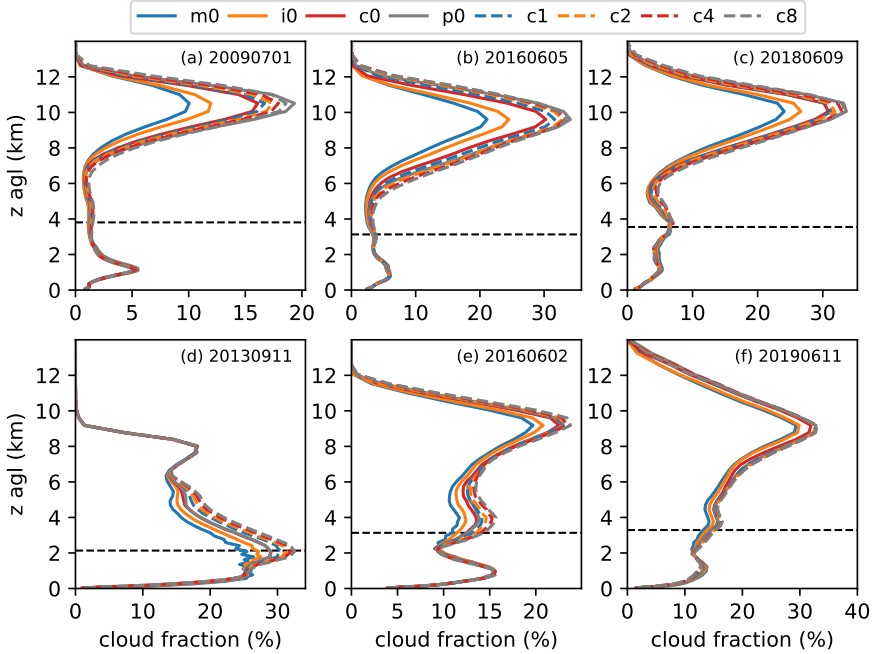

**Figure 7.** Domain-averaged profiles of cloud fraction for weak forcing (top) and strong forcing (bottom). The dashed black lines indicate the height of the freezing level.

as a result of different rain intensities, low-level wind convergence is modified. Similar findings are valid for 500-hPa relative humidity or wind shear parameters. We therefore conclude that the inclusion of microphysical uncertainties by different CCN

concentrations and shape parameters only has implications on environmental variables after precipitation onset. As already mentioned earlier, however, the modified environmental conditions can be important for convective processes at later times.

To assess the relevance of our microphysical perturbations on the vertical cloud structure, we computed mean profiles of the cloud fraction for all cases and sensitivity experiments (Fig. 7). All the weak forcing cases show rather similar profiles below the freezing level. It is only above the freezing level, that the cloud fraction is influenced by different CCN concentrations or

shape parameters with the strongest difference occurring at a height of 10 km agl. For all cases, the cloud fraction increases with increasing CCN concentration. The runs with variations in the shape parameter all lie in between the reference run (c0) and the one with continental polluted aerosol assumption (p0). The maximum difference reaches almost a factor of 2 (from 10% to 19 % on 1 July 2009) which has important implications on the incoming radiation and the energy budget at the ground.

In general, the cases with strong synoptic forcing show a similar behavior, with two exceptions: (i) cloud fraction is altered

already below the freezing level and (ii) all CDSD runs show higher or similar cloud fractions than the p0-run. In addition, the range of simulated differences is smaller than for the cases with weak forcing. This smaller response to the applied microphysical uncertainties could be related to the different cloud and precipitation structure of this weather regime. Here, larger cloud clusters are simulated in comparison to the more scattered and isolated convection in the weak forcing cases (see Fig. 4). These





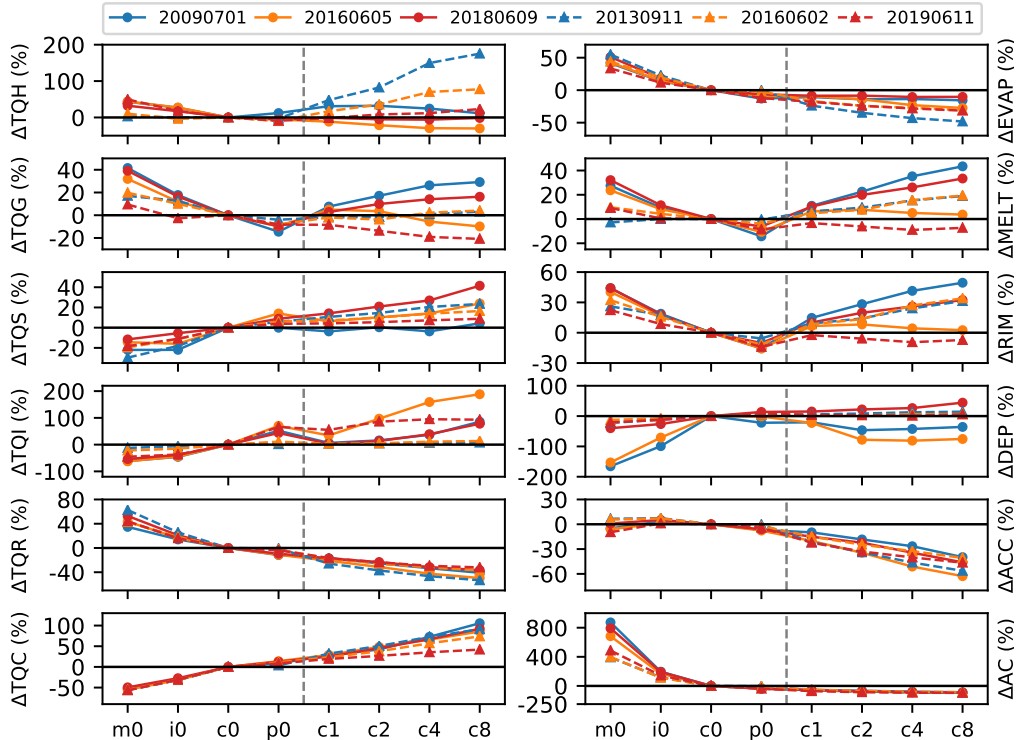

**Figure 8.** Spatio-temporal averages of percentage deviations from the reference run of total cloud water (TQC), rain water (TQR), ice (TQI), snow (TQS), graupel (TQG), and hail (TQH) amounts (left) and of autoconversion (AC), accretion (ACC), deposition (DEP), riming (RIM), melting (MELT), and evaporation (EVAP, right). Weak forcing cases have solid lines, strong forcing cases have dashed ones.

larger cloud clusters seem less susceptible to our modifications which is also apparent in the weaker sensitivity of precipitation
totals for that weather regime. The case of 11 September 2013 (Fig. 7d) shows a sensitivity to our perturbations already from a
height level of 1 km with maximum differences simulated at the freezing level. The height of the freezing level is with 2.1 km
agl the lowest one of our cases and the clouds do not reach so high levels as do the remaining days. The larger fraction of strat-
iform precipitation with a probably higher relative contribution of warm-rain processes might be the reason for this behavior.
This will be analyzed next with the analysis of hydrometeor contents and microphysical process rates.

## 3.3 Total hydrometeor content and microphysical process rates

To further elucidate the impacts of microphysical uncertainties on clouds and precipitation and the processes involved, we now
analyze deviations from the reference run of the total cloud water (QC), rain water (QR), ice (QI), snow (QS), graupel (QG),
hail (QH), and several microphysical process rates (Fig. 8). In agreement with previous findings with the COSMO model (e.g.
Schneider et al., 2019), we find a systematic increase in QC with increasing CCN concentration. This trend is continued with
continental CCN concentration and increased shape parameters (run c1–c8). The sensitivity of QC is quite distinct, ranging





from 56% reduction in a clean environment to more than 105% increase. The total rain water content, on the other hand, is systematically decreasing with increasing CCN concentrations and larger shape parameters. The percentage deviations are smaller than for QC, ranging from a 63% increase to 50% decrease. There is not much of a difference for cases with weak or strong synoptic-scale forcing. As both the CCN increase and larger shape parameters narrow the CDSD, a reduction of the

collision-coalescence process is expected. This is also obvious in our model runs. Compared to the reference run, a further increase in CCN or larger shape parameter systematically decrease the autoconversion of cloud water to rain by up to -95% for shape parameter of 8. For maritime conditions, however, a huge increase in autoconversion between 392% and 873% is simulated. The accretion is also reduced for continental polluted CCN assumptions and narrower CDSD (runs c1–c8), but less intense than autoconversion. As in the study of Barthlott and Hoose (2018), the maximum accretion rates are found in the

respective intermediate runs and for maritime CCN concentrations, positive and negative deviations exist depending on the case.

The ICON model simulates an increase in cloud ice ranging from -50% to +200%, probably due to the larger water load at higher levels caused by the reduced warm-rain process. However, two of the strong forcing cases show much smaller changes in cloud ice from -22% to +13%. The snow content reveals a similar behavior as the cloud ice, but the response is much

smaller. Except the intermediate CCN concentration of the strong forcing case of 11 June 2019, the graupel contents of all runs decrease with increasing CCN. The response of graupel to narrower CDSD shows contrasting results: some of the days show a graupel increase, whereas other reveal a decrease. The same is valid for the total hail content. However, hail amounts are rather small, and only small changes can lead to large percentage deviations. The run with the lowest hail amounts is the one with the strongest hail increase (11 September 2013), which had embedded convection in a frontal rainband. Vapor deposition

mostly shows a systematic decrease with increasing CCN concentrations and also larger shape parameters, indicating a stronger Wegener-Bergeron-Findeisen process that consumes more water vapor to form ice. Two of the cases analyzed (weak forcing cases of 1 July 2009 and 5 June 2016), reveal larger negative deviations in general and maximum deposition rates in the reference run. The reason for that remains unclear, but could be related to the fact that the vertically integrated values also contain negative contributions from sublimation. Especially for these two cases, sublimation at lower levers and deposition at

higher levels have similar values which leads to only small vertical integrals. For the remaining cases, deposition is much higher than sublimation, and the resulting integrals are larger and more significant. The riming of ice particles (ice, snow, graupel, and hail) with supercooled liquid water is decreasing with increasing CCN concentration. As stated by Cui et al. (2011), the size of the graupel particles, the concentration and size distribution of drops, and the collision kernel determine their growth rate by riming. They argue that in high aerosol concentrations, the concentration of cloud droplets is high, but the smaller graupel

particles lead to a smaller graupel–drop collision kernel. This then leads to reduction in riming, which is also true in our simulations and previous studies with the COSMO model (Barthlott and Hoose, 2018). For two of our cases, the larger shape parameter and narrower CDSD leads to slight reduction and increase in riming (5 June 2016 and 11 June 2019). The remaining four cases, however, reveal a strong systematic riming increase with larger shape parameters. The maximum increase with very narrow CDSD (run c8) is even higher as the maritime run with broad CDSD. Those cases reveal a higher cloud droplet number

concentration than in the reference run. This higher number concentration together with the strong increase in total cloud water





could offset the effect of smaller graupel particle size on the collision kernel. However, we must state that we analyze total riming which includes riming on all ice particles and not solely graupel particles.

The response of melting is similar to the one of riming: melting generally decreases with increasing CCN concentrations and increases with larger shape parameters for most of the analyzed cases. However, melting on 11 June 2019 is smaller for 305 larger shape parameters than the reference run. Also, melting on 11 September 2013 is rather insensitive to the aerosol load. Convection on that day was weaker and did not reach such high levels than the remaining days (see cloud fraction profiles in Fig. 7). As already documented by Rasmussen and Heymsfield (1987), the melting of graupel and hailstones is significantly affected by the initial hydrometeor density, their initial size as well as the temperature and humidity profile. A size distribution with smaller and lighter frozen hydrometeors produces the largest melting rates. As neither the graupel nor the hail size 310 distribution shows more smaller particles in maritime conditions or with narrower CDSD (not shown), the largest melting rates are supposed to be a result of the larger graupel and hail content at those runs. This assumption is further supported by the reduced melting on 11 June 2019 in the more narrow CDSD, because also the graupel content is smaller for these runs.

The falling precipitation particles can significantly modify the environmental conditions which feeds back to other microphysical processes. Another important parameter for surface precipitation is the evaporation of rain drops. Our results show 315 largest evaporation rates for clean conditions. Increasing the CCN concentration always leads to smaller evaporation rates. This trend is continued for most cases also with larger shape parameters, whereas two cases show only a small reduction. The reduced evaporation in more polluted conditions is in agreement with previous studies (e. g. Altaratz et al., 2008; Storer et al., 2010; May et al., 2011). As the rain drop size distribution shifts to populations of rain drops that are fewer in number, but larger in size with increasing CCN concentration (Barthlott et al., 2017), evaporation is reduced due to the smaller surface area of 320 large rain drops relative to their volume. The stronger evaporation in clean conditions is resulting from the higher number of small rain drops. The further reduction of rain water evaporation with larger values of the shape parameter is a consequence of the rain drop size distributions which shift to larger sizes than in the CCN sensitivity runs. As the portion of smaller droplets in the population is declining further, the evaporation is declining as well. In addition, larger evaporation rates are also produced by the larger rain water content available for evaporation.

Although not being shown here, we also analyzed the cloud droplet nucleation by CCN activation and the condensation rate from the saturation adjustment. We find that cloud water formation is dominated to a large extent by the saturation adjustment. As expected, the activation of aerosol particles is strongly reduced in maritime conditions and increases with increasing CCN concentration. The nucleation for larger values of the shape parameter is somewhat lower than the one from the respective reference run, but the sensitivity to the shape parameter is rather low.

**3.4 Role of evaporation on surface precipitation amounts**

We showed in the previous section that changes in the CCN concentration and modifications of the shape parameter induce strong percentage deviations in vertically integrated hydrometeor contents and microphysical process rates. It is therefore of interest to determine the reasons why the big changes in cloud water from -56% to 105% and changes in rain water from +63% to -50% only have a smaller impact on precipitation amounts which show deviations of +13% to -16% only. We therefore





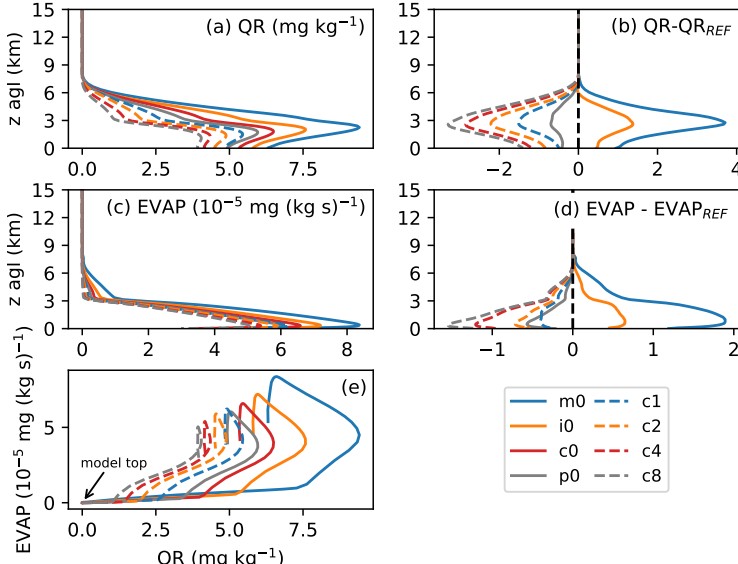

**Figure 9.** Vertical profiles of spatio-temporal averages of rain water content (QR) and evaporation (EVAP) on 1 July 2009. Panel (e) displays the correlation between EVAP and QR throughout the atmosphere with the origin corresponding to the highest model level.

analyze vertical profiles of rain water and rain evaporation in Fig. 9. As the other days generally show similar characteristics, we restrict this analysis to one case study only (5 June 2016). In agreement with previous findings with the COSMO model (Barthlott and Hoose, 2018), the rain water content is much higher in clean conditions. With increasing CCN concentration and larger shape parameters, a QR reduction is simulated and the height of the rain water maximum decreases. The difference plot in Fig. 9b shows this systematic dependance even clearer. Below the level of maximum QR, the profiles converge towards the

ground, leading to a smaller precipitation deviation at the ground. The profile of rain water evaporation displayed in Fig. 9c and d reveal a stronger evaporation for clean conditions. As already mentioned in the previous section, this can be explained by the higher number of small rain drops in that scenario. The evaporation is not strong enough to completely erase the sensitivity of surface precipitation, but we hypothesize that the relative humidity at low levels is decisive for the strength of the precipitation response. The decisive role of dry layers for aerosol effects on deep convective clouds was already documented by, e.g., Grant

and van den Heever (2015). However, we do not observe a linear relationship of evaporation rates with rain water content. In other words, evaporation is not increasing just because there is more rain water. As obvious from Fig. 9e, there is a strong systematic dependance of the evaporation–rain water relationship on CCN concentrations and shape parameter. The strongest sensitivity occurs at maximum rain water contents. Going further down towards the ground, evaporation still increases until the lower parts of the boundary layer are reached and rain water is almost insensitive to evaporation. Furthermore, evaporation

also depends on the size of the clouds itself. The effect of evaporation and entrainment of drier environmental air is larger for populations with smaller sizes than larger cloud systems simulated e.g. in our strong forcing cases.





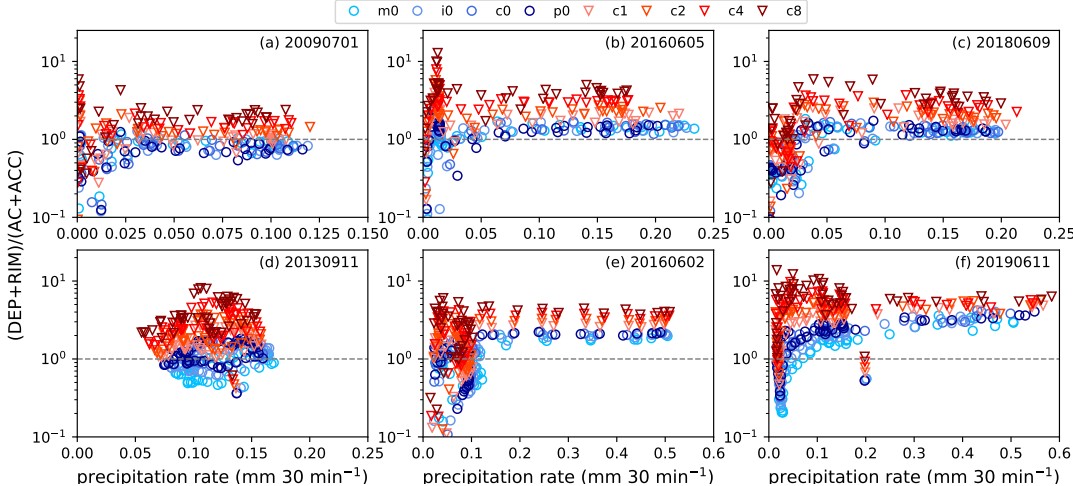

**Figure 10.** Ratio of cold-rain formation (deposition DEP and riming RIM) to warm-rain formation (autoconversion AC and accretion ACC) for weak synoptic forcing (a–c) and strong synoptic forcing (d–f) as a function of mean precipitation rate. Note the different abscissa ranges.

### 3.5 Relative importance of warm-rain and cold-rain process

In section 3.3, we analyzed percentage differences of several microphysical process rates. As neither their absolute values nor their relative contribution to warm and cold-rain processes were addressed, we now analyze the ratio of the sum of vapor

deposition and riming to the sum of autoconversion and accretion as a function of mean precipitation rate (Fig. 10). We find a dominant cold-rain contribution for most of the cases. Only for weaker precipitation rates, the ratio can be smaller than 1, indicating a dominant contribution of the warm-rain process. The weak forcing case of 1 July 2009 (Fig. 10a) reveals the smallest rain intensities of all analyzed cases. For that day, most of the CCN sensitivity runs possess a dominant warm-rain process, whereas most of the CDSD sensitivity runs have a dominant cold-rain process. There is not a clear systematic

dependance of the relative contribution on the CCN concentration. Only for two cases (11 September 2013 and 11 June 2019), the contribution of the cold-rain process increases with increasing CCN concentration. The response of the CDSD sensitivity runs give a clearer picture. The higher the shape parameter, the larger is the cold-rain contribution. Interestingly, the ratios of the CDSD sensitivity runs converge to larger values with increasing precipitation rates than do the ones of the CCN sensitivity runs. This implies that the narrowing of the CDSD not only decreases the absolute values of autoconversion and accretion (see

Fig. 8), but also decreases the relative role of the warm-rain process to rain formation in general, independent of the prevailing weather regime.

Beside the relative role of the warm and cold-rain processes for precipitation amounts, it is also of interest to study the magnitude of the two warm-rain processes autoconversion and accretion and their relative contribution to the warm-rain formation. From Fig. 8, we have seen that autoconversion decreased with increasing CCN concentration and larger values of the

shape parameter. For most of the analyzed cases, accretion showed a similar trend for narrower CDSD, but with smaller mag-





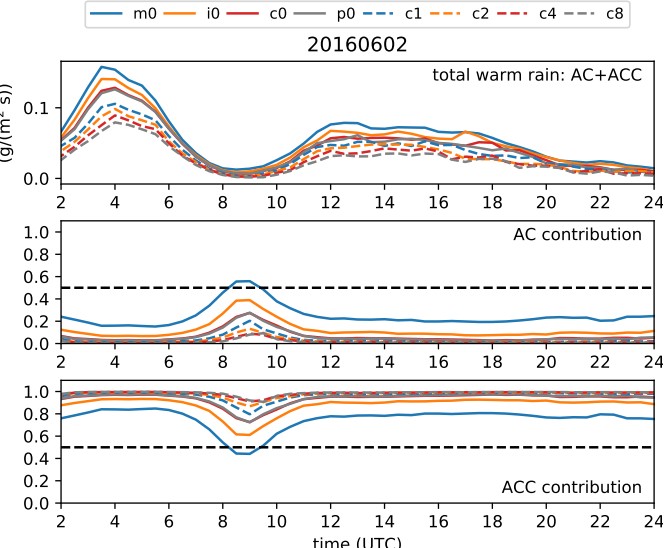

**Figure 11.** Sum of warm-rain processes autoconversion AC and accretion ACC (top), ratio of AC contribution to warm-rain formation (middle), and ratio of ACC contribution to warm-rain formation (bottom).

nitude. Maximum accretion rates were simulated for intermediate CCN concentrations. For the sake of brevity, the temporal evolution of the total warm-rain process and their contributing parts is given for one case only (2 June 2016, Fig. 11). We find that the strength of the warm-rain process is decreasing with increasing CCN concentration. However, the reference run (c0) and the continental polluted run (p0) are almost identical and no further warm-rain reduction is present. The runs with larger

shape parameters reveal a further systematic reduction in the total warm-rain process. Overall, the contribution of accretion to the warm-rain processes outweighs the one from autoconversion. Only in the initial cloud formation phase between 0800– 1000 UTC, the contribution of autoconversion becomes larger and even dominant for a short period of time in the maritime run (m0). In that time period, the ratio becomes smaller for higher CCN concentrations and larger shape parameters. For the remaining times of the day, when also precipitation intensities are high, there is only a decreasing AC contribution (i.e. increas-

ing ACC contribution) for larger aerosol loads. The contribution of both processes remains more or less identical for shape parameter variations. Other days show generally a similar behavior with dominating accretion to the warm-rain process.

### 3.6 Impact on particle size and cloud optical depth

Both the CCN concentration and the shape parameter $\nu$ have an impact on the cloud droplet size distribution. With our simulation setup, we can investigate how large the effects are and which of our sensitivities has the largest relative impact. At first,

we calculated CDSDs for all cloudy grid points (liquid water content threshold of 0.01 $\mathrm{g\,m^{-3}}$) and determined the particle diameter of the dominant value of each CDSD. The response of this dominant cloud droplet diameter at a height of 3 km agl shows an opposing trend in our sensitivity studies (Fig. 12a). The increase in aerosol load leads to more numerous, but smaller



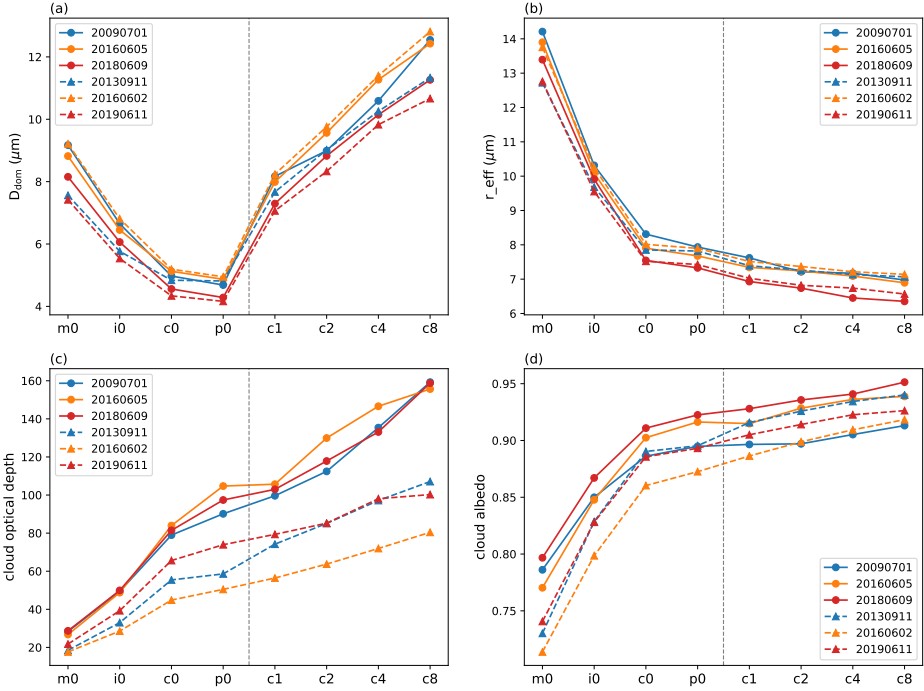

**Figure 12.** Spatio-temporal averages of the dominant diameter in the cloud droplet size distribution $D_{\mathrm{dom}}$ (a), effective radius $r_{\mathrm{eff}}$ (b), cloud optical depth $\tau_c$ (c), and synthetic cloud albedo (d).

droplets and the maximum of the CDSD shifts to smaller values. In addition, the portion of larger cloud droplets decreases (not shown). On the opposite, increasing the shape parameter leads to a systematic increase in the size of the dominant cloud

droplets. By just increasing the shape parameter from 0 to 1, the dominant values in continental CCN amounts are as large as in clean air (run m0).

To better quantify the differences in the entire CDSD, we computed the effective radius as the third moment of the cloud droplet size distribution over the second moment:

$$r_{\mathrm{eff}} = \frac{\int r^3 n(r)\mathrm{d}r}{\int r^2 n(r)\mathrm{d}r}, \tag{2}$$

where $r$ is the droplet radius and $n(r)$ the cloud droplet size distribution. In agreement with findings of, e.g., Peng et al. (2002), the effective radius decreases with increasing aerosol load, similarly as the dominant cloud droplet size $D_{\mathrm{dom}}$. Whereas $D_{\mathrm{dom}}$ then increases with larger shape parameters, the effective radius continues to decrease moderately. This is the result of the narrower size distribution which also reduces the portion of larger cloud droplets. Such a behavior was also documented for idealized test cases (Morrison and Grabowski, 2007; Igel and van den Heever, 2017a). The cloud droplet growth in polluted

clouds is limited by the competition of the available water vapor by the more numerous cloud droplets inside the cloud, which results in smaller effective radii. The effective radius and the cloud liquid water path (LWP) can then be used to approximate





the cloud optical thickness $\tau_c$ and the synthetic cloud albedo $A$:

$$\tau_c = \frac{3\text{LWP}}{2\rho_L r_{\text{eff}}} \tag{3}$$

$$A = \frac{\tau_c}{6.8 + \tau_c}, \tag{4}$$

where $\rho_L$ is the liquid water density. As can be seen in Fig. 12c, cloud optical depth is systematically increasing with increasing CCN concentration and larger shape parameters. The fact that the cloud optical depth increases in a similar magnitude for larger shape parameters as for increasing CCN concentrations can only partly be explained by the decrease of the effective radius, which is considerably smaller in runs c1–c8 than in runs m0–p0. The strong increase in the cloud liquid water path for larger shape parameters (see Fig. 8) seems to be the main contributor for the increase in the cloud albedo for narrower CDSD. An

increase of the CCN concentration will generally lead to an increase in the cloud droplet number concentration. Combined with the increase in the cloud liquid water path, an increase in cloud albedo is expected. There is also a significant increase in the cloud albedo with higher aerosol loads, but the further increase with larger shape parameters is lower than for the cloud optical depth. A change in cloud optical depth or cloud albedo has important implications for the energy and radiation balance at the ground. To quantify that, we determined the range of the net radiation $Q$ at the time of each maximum around

noon (not shown). The ICON model simulates differences of the sensitivity runs for the six cases ranging from 5–41 $\text{W m}^{-2}$ and a mean value for all days of 22 $\text{W m}^{-2}$. As the total cloud cover only changes by 0.4% on average, these changes can mostly be attributed to the different cloud optical properties. Together with the ground heat flux, the differences in the net radiation determine the available energy at the ground for the sensible and latent heat fluxes. This demonstrates the importance of both CCN assumptions and shape of the CDSD also for accurately simulating the processes in the boundary layer which

are important for convection initiation. To sum up, changes in the dominant cloud droplet size and the cloud optical depth are of similar magnitude in the CCN sensitivity runs and the shape parameter sensitivity runs, whereas the effective radius and synthetic cloud albedo show the strongest response for varying CCN concentrations and a smaller sensitivity to variations of the shape parameter. The strong increase of the cloud optical depth shows that the choice of the shape parameter is indeed highly relevant for determining cloud radiative characteristics.

Furthermore, the terminal velocity of cloud drops and rain drops used in numerical models can significantly affect weather predictions (e.g. Ong et al., 2021). As cloud droplets are smaller than 50 µm in radius and thus have no appreciable terminal fall speed relative to the airflow, cloud droplet sedimentation is not included in the ICON model. The terminal velocity of rain drops, however, is important for the rate of removal of liquid water from the atmosphere and depends on the mean droplet diameter, which itself is impacted by the different CCN assumptions and CDSD shape parameters. Moreover, the evaporation

rate of rain droplets will also depend on the terminal fall velocity. If melting particles fall slower, the time for melting increases, which consequently reduces their mass and terminal velocity. Thus, the different cloud droplet size distributions also have a secondary impact on precipitation rates on the ground by influencing the rain droplet size distribution.





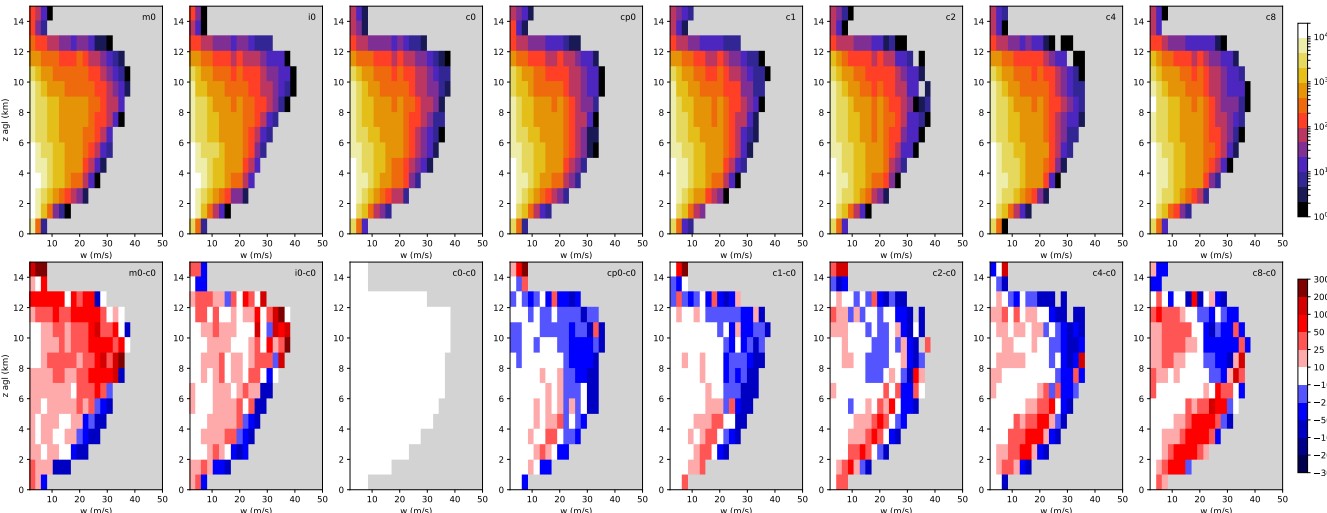

**Figure 13.** Frequency of updraft grid points as a function of height and updraft magnitude (top) and frequency difference in % with respect to the reference run (bottom) on 9 June 2018.

### 3.7 Effects on updrafts

Finally, we will investigate the impact of different aerosol loads and CDSD assumptions on convective cloud updrafts. As an
example, we assess the frequency of updraft grid points as a function of height and updraft magnitude for the case of 9 June 2018
(Fig. 13). Here, only grid points with updrafts larger than $2\,\mathrm{m\,s^{-1}}$ are considered for hourly model output throughout the entire
day. As obvious from percentage deviations to the reference run given in the lower panel of Fig. 13, the number of convective
updrafts decreases with increasing CCN concentrations for most of the updraft magnitudes, indicating a negative aerosol effect.
As was already pointed out for COSMO simulations using the same microphysical scheme, this may be attributed to the fact
that updrafts in polluted conditions contain more water and are therefore less buoyant (Barthlott and Hoose, 2018). Lebo and
Seinfeld (2011) stated that aerosol-induced effects are impacted by the balance between latent heating and the increase in
condensed water aloft with opposing effects on the buoyancy. A negative aerosol effect was also found by Seifert and Beheng
(2006b) for ordinary cells. They argue that in a clean environment, less water freezes and the freezing also occurs at lower
levels, which fosters the dynamics more efficiently. Only for the thin range with largest velocity classes below 5 km agl, the
CCN runs simulate less frequent stronger velocities than the reference run.

The increase of the shape parameter is more complex and shows two main characteristics: (i) below 6–7 km agl there is an
increase in most of the updraft classes except the extreme classes which occur less often than in the reference run. (ii) above
this level, updrafts less than $20\,\mathrm{m\,s^{-1}}$ trend to increase with larger shape parameters, whereas the stronger updrafts occur
less often. The features found for this case are generally also present for the other days analyzed in this study, but some of
them in weaker expression. These findings demonstrate the large impact of our microphysical uncertainties on the dynamics
of convective weather regimes as some configurations produce significantly more and/or larger convective updrafts. Moreover,



more feedbacks such as the formation of stronger cold pools modifying the life cycle of multicell storms or the secondary initiation of convection at cold pool boundaries are presumably also involved, all of which could not be investigated here.

## 4  Summary and conclusions

The purpose of this study was to investigate the range of uncertainties for convective-scale predictability resulting from varying aerosol concentrations and different shape parameters of the cloud droplet size distribution and how the respective sensitivities compare to each other. To this end, we performed convection-resolving simulations with the ICON model for six real-case events over Germany classified into weak and strong synoptic-scale forcing. For each of the investigated cases, we conducted a set with four different CCN concentrations using a reference shape parameter ($\nu = 0$) and another set with increasing shape

parameters ($\nu = 1, 2, 4, 8$) using the reference CCN concentration.

With respect to the reference run, we find a stronger precipitation response for weakly-forced cases than for the strongly-forced cases. Integrated over the domain of Germany, precipitation totals systematically decrease with higher aerosol loads. The fact that the ICON model simulates a negative aerosol effect for all cases irrespective of the prevailing weather regime is remarkable and in contrast to previous simulations with the COSMO model, which produced an invigoration of convection at

least for some of the weakly-forced cases (Barthlott and Hoose, 2018; Schneider et al., 2019; Keil et al., 2019). Whether this difference is case-dependant or caused by a different implementation of the double-moment scheme in the ICON model (e.g. saturation adjustment before and after the microphysics) cannot be answered here and is left for future work. The narrowing of the cloud droplet size distribution when using larger shape parameters also reduces the precipitation amounts compared to the reference run as well, for most cases even systematically. An important finding is the fact that an increase in the shape parameter

can produce almost as large a variation in precipitation totals as a CCN increase from maritime to polluted conditions. Thus, the shape parameter may be an important parameter in the context of aerosol-cloud interactions, as was already pointed out for idealized simulations by Igel and van den Heever (2017c).

The percentage range of precipitation deviations (i.e. difference between maximum increase or decrease) reaches values larger than 20% for the CCN sensitivities and 16% for the shape parameter sensitivities for individual days. On average over

all days, however, the mean precipitation sensitivity to CCN variations (12%) is larger than for the shape parameter (7%). These large values demonstrate the importance of these microphysical uncertainties for quantitative precipitation forecasting.

For the evaluation domain of Germany, the timing of convective precipitation is insensitive to the aerosol load or the assumed shape of the CDSD. Thus, the differences in precipitation totals are solely generated by weaker or stronger rain intensities. On average, 85% of the 30-min intervals of the model run with maritime CCN and 79% of the one with intermediate CCN

concentration (both with reference shape parameter) have larger mean precipitation rates as the reference run. The model runs with larger shape parameters, on the other hand, show weaker rain intensities for most of the time. Interestingly, one of the cases revealed a turning point from weaker to stronger rain intensities of the CDSD sensitivity runs than the reference run. The weaker rain intensities in the developing phase of convective clouds consume less CAPE and consequently allow for a higher convective instability later on in the evening which produces stronger convection at that time. This highlights the importance



of the interaction of the convective development with its thermodynamical environment which could be considered as another "lifetime effect", similar to the one proposed by Albrecht (1989), but more related to the thermodynamics and prolonged instability. We therefore conclude that this interaction can be an important point for other cases as well as the total precipitation amount can depend on whether (i) such a turning point exists and (ii) if it occurs early enough to have a major impact on precipitation totals. Our results show that the simulation of convective precipitation involves complex interactions between

thermodynamic and microphysical processes.

The analysis of the vertically integrated hydrometeor contents shows a large systematic increase in total cloud water content with increasing CCN concentrations and more narrow CDSDs together with a reduction in the total rain water content. We could attribute this to a systematic reduction of autoconversion with increasing CCN concentration and shape parameter. Whereas cloud ice and snow generally show an increasing trend with higher aerosol loads and narrower size distribution, graupel and

hail only show a systematic reduction in the CCN sensitivity experiments due to weaker riming. For higher shape parameters, the graupel and hail response is not systematic and case-dependant. It must be stated that hail contents are low for some of the days and that differences between small values are not reliable and general conclusions cannot be drawn. The big impact of our microphysical uncertainties on the total rain water content ranging from +63% to -50% does not impact the precipitation totals at the ground in the same magnitude. The reason is the evaporation of rain drops at lower levels, which is considered to be a

key process in determining the magnitude and sign of aerosol-cloud-precipitation interactions (e.g. Tao et al., 2007; Grant and van den Heever, 2015; Barthlott et al., 2017). The larger evaporation is not solely induced by the larger rain water contents, but also related to the particle size distribution with a higher number of small rain drops.

Our findings also show a dominating cold-rain process for all cases, whereas the warm-rain process dominates the formation of rain water at weaker rain intensities only. However, there is not a clear systematic dependance of the relative contribution

on the CCN concentration. For larger shape parameters, however, the contribution of the cold-rain processes increase as well. Interestingly, the ratios of the CDSD sensitivity runs converge to larger values with increasing precipitation rates than the CCN sensitivity runs. This implies that the narrowing of the CDSD not only decreases the absolute values of autoconversion and accretion, but also decreases the relative role of the warm-rain process to rain formation in general, independent of the prevailing weather regime. We have also seen that the contribution of accretion dominates the one from autoconversion, only at

initial cloud formation, autoconversion can be larger than accretion for a short period. For autoconversion, changing the CCN concentration leads to a much larger response than does the change in the shape parameter, whereas the opposite is true for the warm-rain dominating accretion. This is related to the fact that clouds in polluted conditions consist of more droplets that coalesce into rain drops less effectively, but the narrowing of the CDSD further reduces the amount of large droplets which seems to have a larger effect on accretion in general.

Our findings also highlight important impacts on the optical properties of the clouds. Whereas the maximum in the CDSD shifts to smaller values when the aerosol load is higher, it increases with larger shape parameters. By just increasing the shape parameter from 0 to 1, the dominant values in continental CCN amounts are as large as in clean air. The effective radius, however, decreases with increasing aerosol load and, to a lesser extent, also with increasing shape parameters as a result of the narrower size distribution which also reduces the portion of larger cloud droplets. We further find a strong systematic increase



in the cloud optical depth with increasing CCN concentrations and larger shape parameters. The synthetic cloud albedo also shows a positive relation to the aerosol load and, to a lesser extent, also to the shape parameter. Especially, the strong increase of the cloud optical depth shows that the choice of the shape parameter is indeed highly relevant for determining cloud radiative characteristics.

Furthermore, our simulations did not reveal an invigoration of convective clouds, a phenomenon also present in COSMO

simulations using the same double-moment scheme used in this study. This could be influenced by the saturation adjustment scheme to treat condensational growth (Barthlott and Hoose, 2018). As stated by Lebo et al. (2012), the potential of a CCN increase to increase buoyancy at mid- to upper levels could be reduced as this technique primarily enhances condensation and latent heating at lower levels.

These findings demonstrate that both, CCN assumptions and the shape parameter, are important for quantitative precipita-

tion forecasting and should be carefully chosen if double-moment schemes are used for modeling aerosol-cloud interactions. The inclusion of microphysical uncertainties by disturbing the shape parameters in ensemble forecasting is therefore very promising. It remains open how aerosol effects are modulated in simulations with different shape parameters. Further work is needed to determine the effects of the shape parameter by combined sensitivity analyses in which CCN concentrations are systematically varied for different values of the shape parameter.

*Data availability.* ICON model output is available on request from the authors.

*Author contributions.* CB and CK developed the project idea and designed the numerical experiments. AZ performed the numerical simulations. CB and AZ conducted the analyses, TM calculated the convective adjustment time scale, and all contributed to the interpretation of the results. CB wrote the paper, with contributions from all co-authors.

*Competing interests.* The authors declare that they have no conflict of interest.

*Acknowledgements.* The research leading to these results has been done within the subproject B3 of the Transregional Collaborative Research Center SFB/TRR 165 "Waves to Weather" (www.wavestoweather.de) funded by the German Research Foundation (DFG). The authors wish to thank the Deutscher Wetterdienst (DWD) for providing the ICON model code. We are also grateful for the provision of initial and boundary data by DWD and the European Centre for Medium-Range Weather Forecasts (ECMWF). This work was performed on the supercomputer ForHLR funded by the Ministry of Science, Research and the Arts Baden-Württemberg and by the Federal Ministry of Education and

Research.





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
