# Peer review of "Importance of aerosols and shape of the cloud droplet size distribution for convective clouds and precipitation"

_Atmospheric Chemistry and Physics, 2021_

## Referee Comment (RC2)

Review of manuscript no. acp-2021-481: **"Importance of aerosols and shape of the cloud droplet size distribution for convective clouds and precipitation"** by Christian Barthlott, Amirmahdi Zarboo, Takumi Matsunobu, and Christian Keil.

**General**

This paper studies the response of convective cloud systems and their precipitation production to changes in CCN and CDSD size distributions (concentration and shape parameter). Using the ICON model, numerical simulations of different synoptic systems (6 real cases) are conducted for the area covering central Europe. Comparison of the clouds and precipitation properties under different CCN size distributions is done for concluding about aerosol effects on deep convective systems. The simulated results, which are classified to weak and strong synoptic forcing, point on increased total cloud water and decreased total surface rain, with increased CCN concentration and narrower size distribution. The precipitation response is stronger for weakly forced cases. Explanations for these results are suggested by analysis of different hydrometeors types in the simulations and the related formation mechanisms. Less efficient collision-coalescence is demonstrated in more polluted cases (suppressed warm rain formation) and stronger rain evaporation at low levels. The simulated results also show a negative effect of aerosol on the convective intensity meaning there is no convective invigoration. This work examines the interaction of the clouds with their thermodynamic environment as well showing the impact of precipitation on the environmental instability.

To summarize, it is a very interesting and valuable work dealing with an important subject which is still not fully understood.

I have a few comments that should be addressed before publication:

1) This study used a bulk microphysical scheme with a saturation adjustment assumption. This method is limited in its ability to simulate rightly the aerosol effect on warm cloud processes. First, the condensation efficiency cannot be accurately described by a saturation adjustment scheme. It was shown that the supersaturation values in clouds depend heavily on the aerosol loading (Pinsky et al., 2013, Seiki and Nakajima, 2014, Dagan et al., 2015). This major effect is neglected in this work. Another major effect is the aerosol impact on the drops' effective terminal velocity (Koren et al., 2015). A bulk scheme is limited in its ability to describe the full

range of terminal velocities and so it neglects this major effect too. The authors should regard this major issue and the limitations of the method used here for this type of study should be discussed in more details.

2) This study examines the effects of changes in CCN and CDSD size distributions on deep convective systems. It examines both warm and cold processes. Nevertheless, there is no description or treatment of changes in IN size distribution. If there are changes in aerosol properties it will affect the IN properties and hence the mixed and cold processes. This issue should be explained in the manuscript regarding the treatment of IN in the model and its consequences on the results.

3) The terminology used in the paper for describing the aerosol loading is very confusing (maritime, continental, polluted). The use of maritime and continental can regard the thermodynamic conditions as well and that why it is confusing. I suggest to change it to clean, intermediate, polluted and highly polluted and to use it in a consistent way throughout the paper.

4) In order to validate the model simulations there is a need to compare it to measurements. I suggest to add a figure which is similar to fig. 4 that will present observed accumulated rain or some other cloud properties for the 6 cases. This will enable estimation of the validity of the simulated results.

5) Fig. 6: The meaning of the shading is not explained in the figure caption so the figure is unclear. It should be added.

6) The idea of considering the interaction of the convective clouds and the available instability as a type of "lifetime effect" is problematic as it treats a whole cloud system and not a single cloud (as the lifetime effect). This idea should be examined again and any way it should be explained better.

7) Fig. 8: I suggest to present mean vertical profiles of the different types of hydrometeors and cloud processes instead of the way it is presented now in figure 8 (similarly to fig. 7). The suggested way of presentation will connect better to fig. 7 and will help to present a full picture of the explanations.

**The relevant papers:**

1. Pinsky, M., Mazin, I., Korolev, A., and Khain, A. (2013): Supersaturation and diffusional droplet growth in liquid clouds, J. Atmos. Sci., 70, 2778‑2793.

2. Seiki, T. and Nakajima, T. (2014): Aerosol effects of the condensation process on a convective cloud simulation, J. Atmos. Sci., 71, 833‑ 853.

3. Koren I., Altaratz O. and Dagan G. (2015): Aerosol effect on the mobility of cloud droplets. Environmental Research Letters. 10, 10, 104011.

4. Dagan G., Koren I. & Altaratz O. (2015): Competition between core and periphery-based processes in warm convective clouds - from invigoration to suppression. Atmospheric Chemistry and Physics. 15, 5, p. 2749-2760

---

## Author Comment (AC1)

**Responses to the reviewers**

Importance of aerosols and shape of the cloud droplet size distribution for convective clouds and precipitation

by C. Barthlott, A. Zarboo, T. Matsunobu, and C. Keil                      September 29, 2021
* * *
We thank both reviewers for reading the manuscript and providing detailed comments. We have carefully considered all comments and changed the manuscript accordingly. Please find below our responses in blue.

**Reviewer 1**

This study attempts to shed more light onto the effects of CCN and shape parameter assumptions on cloud and precipitation properties in the context of NWP, and how the effects differ between differently forced cases. It provides interesting and valuable findings to the existing knowledge and I would like to see it published eventually, since these two microphysical properties are often not very well constrained by observations as pointed out by the authors. However, some analysis and general scientific communication need some major revision to address the concerns.

**Major Comments**

1. Definition of shape parameter and gamma distribution: When talking about the shape parameter in the introduction, the authors implicitly switched between shape parameter for the size distribution and that for the mass distribution (given by Eq. 1), which has a difference of 2. Normally, the size distribution collapses into an exponential distribution for $\nu=0$, but it is not the case for Fig. 1, which can cause some confusion. Note that nothing is explicitly incorrect, just confusing. This difference deserves to be explained in greater detail in the introduction. Also, when the authors cite that typical values of the shape parameter are 0-14, they are citing studies which calculated the shape parameter for a size distribution. To apply these values to a mass distribution instead, 2 should be subtracted. In other words, the authors cite that a typical range is 0-14 and then test a range of 2-10 (in terms of the size distribution, it's 0-8 in terms of the mass distribution). This may be important because previous studies (e.g. Igel and van den Heever) have found the greatest sensitivity at low values of the shape parameter. The authors may want to test a mass distribution shape parameter of -2 corresponding to a size distribution shape parameter of 0.

   Our equation (1) is based on the particle mass $x$, but can easily be transferred to particle sizes using a power law for the diameter-mass relation. We added the necessary calculation steps in the paper:

   *"Using a power law for the diameter-mass relation $D(x) = ax^b$ ($a = 0.124$ m kg$^{-b}$, $b = 1/3$), we can transform Eq. 1 from mass $x$ to particle diameter $D$:*

   $$f(D)\mathrm{d}D = f(x)\mathrm{d}x \tag{2}$$
   $$f(D) = f(x)/(\mathrm{d}D/\mathrm{d}x) = f(x)/bax^{(b-1)} \tag{3}$$

   *An equivalent way to convert the size distribution from mass to radius (or diameter) for spherical particles is given in Khain et al. (2015):*

   $$f(D) = N_0' D^{\nu'} \exp\left(-\lambda' D^{\mu'}\right) \tag{4}$$

with $N_0' = 3N_0 \left(\frac{\pi}{6}\rho\right)^{\nu+1}$, $\nu' = 3\nu + 2$, $\lambda' = \lambda \left(\frac{\pi}{6}\rho\right)^{\mu}$, and $\mu' = 3\mu$. $N_0$ is the intercept parameter and $\rho$ the bulk hydrometeor density. Both Eqs. 3 and 4 give the same result for the size distribution as a function of the particle diameter."

We also stated that the literature values are based on particle diameter $D$ and added a further comment here:

*Figure 1 presents Gamma size distributions with different shape parameters for a fixed cloud water content (QC) and cloud droplet number concentration (QNC) as a function of particle diameter $D$ (see Eq. 3) using a dispersion parameter $\mu = 0.33$.  It can be seen that increasing the shape parameter narrows the size distribution.*

At the end of section 2.1 (Model description and simulation overview), we added the variable $\nu$ in the text and Tab. 1 to make clear which parameter we modify. Using notation (4) for the CDSD as a function of $D$ and $\nu'$ as shape parameter would mean that our values $\nu = 0, 1, 2, 4, 8$ would become $\nu = 2, 5, 8, 14, 26$ as $\nu' = 3\nu + 2$.

Equation 1 only collapses to the exponential distribution for particle mass $x$ with $\nu = 0$ AND $\mu = 0$. This is not the case for our Figure 1, where the distribution is displayed as a function of particle diameter $D$ and $\mu = 1/3$. Our description of the size distribution also collapses into an exponential distribution for $\nu = 0$ and $\mu = 0$ (Seifert and Beheng, 2006a). We added this information in the text:

*"This function reduces to the classical $\Gamma$-distribution with $\mu = 1$, to the Weibull distribution with $\nu = \mu - 1$, and to the exponential distribution with $\mu = \nu = 0$ as a function of particle mass (Seifert and Beheng (2006a)."*

Furthermore, the reference value of $\nu = 0$ shows already a very broad size distribution with a maximum at small diameters. A further reduction to $-2$ is possible, but the resulting size distribution would be even broader and less realistic. As suggested by the reviewer, we made a model run for 11 June 2019 with a shape parameter of $\nu = -2$ and display the 24-h precipitation amount in Fig. R.1 together with the results of the other runs with varied shape parameters. It can be seen that the shape parameter of -2 leads to a precipitation reduction compared to the reference value of more than -23%, whereas the runs with $\nu = 8$ has a reduction of -8% only. We therefore believe that the values of the shape parameter are well chosen in this study and reflect the observational uncertainties of this parameter reasonably well.

2. In order to help readers understand how the shape parameter influences the simulations, could the authors please briefly discuss which processes in the model make use of this parameter? Autoconversion, accretion, riming, evaporation, radiation, etc?

   We mentioned some of these processes already in the Introduction and now added more information at the end of the model description:

   *"The size distribution of the cloud droplets has a substantial impact on the simulation results, as various microphysical processes such as condensation, evaporation, autoconversion, accretion, riming, and sedimentation depend on it. Since the optical properties of clouds are also influenced, the shape of the CDSD is also important for the radiation and the energy balance at the surface."*

3. The authors devote nearly a page to CAPE and thermodynamics and include summaries of the discussion in the abstract and conclusions. Since this is such an important part, please show some figures to support your conclusions.

   Already when writing the first version of the manuscript, we thought about including the temporal evolution of CAPE, but decided not to include it due to the already high number of figures. Now, we have included this figure together with the lapse rates as new Fig. 7 to support our

[Figure]

Figure R.1: 24-h precipitation amount on 11 June 2019 for continental CCN concentration and different shape parameters (top left: -2, top center: 0, top right: 1, bottom left: 2, bottom center: 4, bottom right: 8)

description in the text. We think that this supports our conclusions.

4. Can the authors discuss why there appears to be virtually no sensitivity to either CCN or the shape parameter in the first 10-12 hours of any simulation, even if precipitation rate is high (e.g. Fig. 6, case 20160602)? Perhaps this is just an artefact of averaging over the entire domain? If averages are taken only where there is precipitation, does this lack of sensitivity disappear?
   The two strong forcing cases of 20130911 and 20160602 show the smallest response in the 24-h amount of the six cases analyzed in this study. The precipitation deviation from the reference run lies between -5% and +4%. The general weaker response of preciptiation to varied CCN concentrations for cases with strong synoptic forcing was also documented by Barthlott and Hoose (2018) or Schneider et al. (2019). As suggested by the reviewer, we calculated the mean precipitation rates for rainy grid points only (Fig. R.2) and find a larger sensitivity for the shape parameter runs and also for the CCN runs in the nighttime maximum for 20160602. We believe that particularly in strong synoptic forcings, clouds behave more like a buffered system and that the sensitivity of surface rainfall to these microphysical uncertainties remains small. We added the following text in the manuscript:

   *"The comparably small spread in precipitation intensities for both the CCN and shape parameter runs during the nighttime precipitation maximum on 2 June 2016 could be explained by the fact that particularly in cases with strong synoptic forcing, clouds act more like a buffered system and the response of precipitation to these microphysical uncertainties remains small."*

**Minor Comments**

1. Line 21-23, "polluted conditions does not lead to an invigoration" is not equivalent to a negative aerosol effect (could also be no effect). I would suggest the authors rephrase this part.
   We rephrased this part, it now reads:

[Figure]

Figure R.2: Domain-averaged precipitation rate for rainy grid points only.

*"By the frequency of updrafts as a function of height, we show a negative aerosol effect on updraft strength, leading to an enervation of deep convection."*

2. Line 27, I think the authors meant lower grid spacing, or higher grid resolution.
   Yes, we meant grid resolution and changed the text accordingly.

3. Line 34, did the authors mean "...are *not* accounted for"?
   No, in that sentence we meant "...are accounted for". The following sentence describes uncertainties from microphysical processes.

4. Line 67, What is $\mu$ assumed to be?
   $\mu$ is the dispersion parameter of the distribution, it is explained in the text. In case of $\mu = 1$ this function reduces to the classical $\Gamma$-distribution, with $\nu = \mu - 1$ to the Weibull distribution and with $\mu = 0, \nu = 0$ to the exponential distribution (Seifert and Beheng, 2006a). This additional information has been added to the introduction.

5. Figure 3 and 4, I appreciate the detailed description of the cases that will be discussed later in the paper.
   Thank you.

6. Line 129, Does RRTM account for the shape parameter of the cloud droplet size distribution?
   Yes, in RRTM the effective radii of cloud droplets are based on mass and number concentrations consistent with the microphysics. The effective radii of cloud ice crystals are calculated based on ice mass concentrations only. Only in the latest model release and the new radiation scheme (ecRAD), which has become operational in April this year, there is a module to compute the effective radius of liquid and ice consistent with 2-moment-microphysics to be used by the radiation module.

7. Table 1, Just to be clear, only the shape parameter of cloud droplets is being changed, correct? Not the shape parameter of any other hydrometeor categories?
   Yes, we only change the shape parameter of the cloud droplet size distribution. We added one

sentence to make that clear:

*"Note that we only change the shape parameter of cloud droplets and not that of other hydrometeor categories."*

8. Line 176, I think the finding that lower CCN concentration leads to precipitation enhancement is very much expected.

   The important finding is the fact that total precipitation systematically decreases for increased CCN concentrations for all cases, independant of the synoptic forcing. The COSMO model with the same microphysics scheme simulated no systematic relationship for weakly forced conditions in earlier research. We have written a separate paragraph on that topic. We also included a citation to a recently published article on invigoration of convective clouds by aerosols by Igel and van den Heever (2021) with this text in the Conclusions:

   *"Despite many efforts with field experiments and state-of-the-art numerical models, the validity of the invigoration effect is still an open question (Altaratz et al., 2014). In a recent study by Igel and van den Heever (2021), theoretical calculations of a new formulation of the moist adiabatic lapse rate that accounts for freezing, supersaturation, and condensate loading were performed. They find that a CCN-induced increase in storm updraft speed, is theoretically possible, but substantially smaller (and oftentimes even negative), than previous calculations suggested."*

   Igel, A. L. and van den Heever, S. C.: Invigoration or Enervation of Convective Clouds by Aerosols?, Geophys. Res. Lett., 48, e2021GL093 804, https://doi.org/10.1029/2021GL093804, 2021.

9. Line 187-188, the word "contrast" is potentially misleading as such contrast only exists due to the arbitrary choice of the "reference run" having continental CCN concentration and a shape parameter of 0. Both the effect from CCN concentration and shape parameter on precipitation seems to be monotonic, hence no real "contrast". Similar languages can also be found in many other places in the manuscript, which need to be revised to avoid being misleading.

   We totally agree with the reviewer and removed that sentence from the manuscript. We also changed one sentence in section 3.3:

   *"The response of graupel to narrower CDSD reveals an opposite behaviour: some of the days show a graupel increase, others a decrease."*

10. Line 211, I'm confused by the word choice "only" as if larger precipitation rate is expected despite the narrower distribution. If the authors chose the word "only" because they are comparing it against the effect from the maritime (or intermediate) CCN case, I should note that these two cases are not comparable as both CCN and shape parameter are different. I do think it is an interesting observation because I would not expect any increase in precipitation rate due to larger shape parameter, but the authors seem to shift the focus from "the effect of CCN and shape parameter on precipitation rates" to a contest of "who is best at increasing the precipitation rate".

    We also did not expect an increase in precipitation rate due to larger shape parameters. But we wanted to quantify for how many times the precipitation rate is above or below the one from the respective reference run. The fact that the shape parameter sensitivity runs have higher precipitation rates in 26–33% of the time, means that lower rain intensities are present for the majority of time. Consequently, the accumulated rain amount is less after 24 h for these runs. We deleted the word "only" from the text and rephrased the following sentence to make that clearer. It now reads:

    *"The shape parameter sensitivity runs have larger precipitation rates in 26–33% of the time. Thus, lower rain intensities than in the respective reference run are present in the majority of time which explains the reduced precipitation totals after 24 h."*

[Figure]

Figure R.3: Ratio of sensitivity to reference run (top) and percentage deviations from the reference run of total cloud water TQC (bottom).

11. Figure 7, have the authors considered showing the same range for the x-axis so that it's easier to compare between cases? I do not claim it is the right way, but it would be more obvious and convincing that cloud fraction in the weaker forcing cases show a stronger influence from different CCN concentrations.
We followed the reviewer's suggestion and modified the x-axis range accordingly.

12. Figure 8, I'd argue that it makes more mathematical sense and is visually more intuitive to plot ratios on a log scale rather than percent deviations. -50% and 200% represent a halving and a doubling, respectively, and are therefore arguably relative changes of the same magnitude. But on a linear scale the 200% appears to be the much larger change. The ratios on a log scale would have the same deviation from 1 and appear equivalent.
Actually, a 200% increase would mean a triplication and not a doubling. We have tested the reviewer's suggestion for total integrated cloud water TQC (Fig. R.3). Although it may mathematically make more sense to plot ratios on a log scale, we believe that the percentage deviations are visually intuitive as well and the percentage numbers may be better suited for understanding the changes with respect to the CCN concentration and shape parameter. We therefore decided to keep Fig. 8 and the entire discussion in section 3.3 based on percentage deviations from the reference run.

13. Line 265-266, can the authors explain why the sensitivity of QC is distinct? It doesn't seem to be outlier compared to other quantities and is very much expected based on the classical theories.
We agree with the reviewer that this behaviour is expected, we just wanted to mention the range of QC deviations. We rephrased the text:
"The sensitivity of QC ranges from 56% reduction in a clean environment to more than 105% increase with continental CCN concentration and narrow CDSD. "

14. Line 280, Please insert "for" between "Except" and "the".
done

15. Line 289, "levers" should be "levels"
    done

16. Line 297, does the author mean just "slight reduction" instead of "slight reduction and increase"?
    We meant a slight reduction for one case and a slight increase for the other. We rephrased the text:
    *"For two of the cases, the larger shape parameter and narrower CDSD leads only to a slight reduction (11 June 2019) and slight increase (5 June 2016) in riming. "*

17. Line 308, Are the effects of initial hydrometeor density and initial size included in the model?
    Yes, all hydrometeor categories have prescribed forms of the initial particle size distribution which evolve as a function of number and mass densities.

18. Figure 10, I would again suggest showing the same range for the x-axis for easier comparison between cases.
    When we use the same x-axis range for all days, details are hard to see for the days with weaker rain intensities (20090701 and 20130911). In particular for the case of 20090701, the behaviour described in the text is not visible anymore. We therefore decided to keep the different axis ranges. A note about the different axis ranges was already included in the figure caption.

19. Line 344-345, This discussion of Grant and van den Heever seems a little out of context since that paper tested midlevel dry layers whereas the authors are discussing RH near the surface.
    We deleted that sentence, but kept a citation to this paper in the introduction.

20. Line 347-348, "The strongest sensitivity occurs ..." Sensitivity of what to what? Please clarify.
    We rephrased this sentence to
    *"The biggest differences resulting from CCN and shape parameter variations occur at maximum rain water contents."*

21. Line 356, If the authors wish to understand the importance of cold rain vs warm rain, why not instead take the ratio of melting to autoconversion and accretion? Perhaps melting and deposition (presumably just net deposition on ice) plus riming are equivalent. Either way, I would explicitly state that the authors are taking the ratio of rain production via ice to the ratio of rain production via warm phase processes.
    We also tested the ratio of melting to autoconversion and accretion and find that the results are merely identical to the ratio using deposition and riming as cold rain process. To make it clear that we mean the ratio of rain production via ice to the rain production via warm phase processes, we rephrased the beginning of section 3.5. It now reads:

    *"As neither their absolute values nor their relative contribution to warm and cold-rain processes were addressed, we now inspect the ratio of rain formation via ice (i.e. the cold-rain contribution as the sum of vapor deposition and riming) to rain formation via warm phase processes (sum of autoconversion and accretion). The analysis of this ratio as a function of mean precipitation rate in Fig. 10 reveals a dominant cold-rain contribution for most of the cases. "*

22. Line 359, I find that people mean many different things by "CDSD sensitivity". I think in this case the authors mean shape parameter sensitivity.
    Throughout the paper, we differentiate between CCN sensitivity and CDSD sensitivity (meaning the shape parameter sensitivity). But as changes in the CCN concentration also affect the cloud droplet size distribution, we agree with the reviewer that it is better to say shape parameter sensitivity instead of CDSD sensitivity. We changed it throughout the entire manuscript. We also changed "CDSD-runs" to "shape parameter-runs".

23. Line 386, can the authors define "dominant value of each CDSD"? Do they mean "modal"?

Yes, the dominant value is the most frequently occurring value in the distribution. We changed it to modal value in the text and also replaced $D_{\mathrm{dom}}$ with $D_{\mathrm{modal}}$.

24. Eq 3, should be the integral of LWC/$r_{\mathrm{eff}}$ constants because reff is also a function of z. Or the authors should specify how they found an average value of $r_{\mathrm{eff}}$ such that it could be pulled out of the integral.

We compute the cloud optical thickness $\tau_c$ after Serrano et al. (2014), who use a simple approximation given by Stephens (1994):

$$\tau_{\mathrm{c}} = \frac{3\mathrm{LWP}}{2\rho_{\mathrm{L}}r_{\mathrm{eff}}} \tag{1}$$

Where LWP is the total vertical liquid path of the cloud, $\rho_{\mathrm{L}}$ is liquid water density and $r_{\mathrm{eff}}$ is droplet effective radius. For computational reasons, we averaged the effective radius over the height levels of 2, 3, 4, 5 km agl.

Serrano, D., Nez, M., Utrillas, M. P., Marn, M. J., Marcos, C., and Martnez-Lozano, J. A.: Effective cloud optical depth for overcast conditions determined with a UV radiometers, Int. J. Climatol., 34, 39393952, https://doi.org/10.1002/joc.3953, 2014.

Stephens GL. 1994. Remote Sensing of the Lower Atmosphere: An Introduction. Oxford Univ Press: New York, NY; 521.

We included the information about averaging $r_{\mathrm{eff}}$ and the reference of Serrano et al. (2014) in the paper.

25. Line 422, I think the small sensitivity is because the reference run is already at a relatively high CCN concentration, hence not a lot of room for $r_{eff}$ to decrease or albedo to increase. The effect of shape parameter could be more noticeable if the reference run has a lower CCN concentration. So it's at least necessary to add some qualification to this sentence.

We do see a strong sensitivity in the dominant cloud droplet size (i.e. the maximum of the CDSD) and the cloud optical depth also for the shape parameter variations, it is only for the effective radius and the cloud albedo that the shape parameter runs show a weaker response than the runs with varied CCN concentrations. So the high CCN concentration of the reference run is not responsible for a general weaker response in the shape parameter runs. However, for most of our results, we find that going from maritime to continental yields a much larger response than from continental to continental polluted conditions. The question if shape parameter variations with other CCN concentrations show a stronger response is currently ongoing work in our group. We added this comment:

*"The weaker response of the effective radius and the cloud albedo to shape parameter variations could be influenced by the fact that the reference run already has a comparatively high CCN concentration. We are currently working on the question of how different shape parameters behave at other CCN concentrations."*

26. Figure 13, typo the ticks on the color bar (some 0s might have stuck out of the figure canvas). I would again suggest plotting the ratio in log scale for more intuitive visualization.

Some of the 0s were missing due to a too small viewport, this has been corrected now. However, we like to stick to our selected visualization form with the logarithmic frequency of updraft grid points and percentage difference as we believe that the necessary features are obvious to the reader. The same form of analysis and display was also performed in a recent model intercomparison effort studying the impacts of varying CCN concentrations on deep convective cloud updrafts by Marinescu et al. (2021). This paper is also cited in our manuscript.

27. For figures that have the different microphysical assumption on the x-axis (Fig. 5, 8, 12), it is probably a better idea to not connect the dots between p0 and c1 since they are from two separate experiment groups and thus not logically related. Doing this will also make the (mostly) monotonic relationship between, for example, precipitation rate and microphysical assumptions more obvious. It may be helpful to repeat the c0 result at the start of the shape parameter test set.

This is a good point, we changed Figs. 5, 8, and 12 in the proposed way. We did not repeat the c0 result at the start of the shape parameter test set, because this would be a repetition and for most illustrations, it is zero anyway.

---

## Author Comment (AC2)

**Responses to the reviewers**

Importance of aerosols and shape of the cloud droplet size distribution for convective clouds and precipitation

by C. Barthlott, A. Zarboo, T. Matsunobu, and C. Keil September 29, 2021

We thank both reviewers for reading the manuscript and providing detailed comments. We have carefully considered all comments and changed the manuscript accordingly. Please find below our responses in blue.

**Reviewer 2**

This paper studies the response of convective cloud systems and their precipitation production to changes in CCN and CDSD size distributions (concentration and shape parameter). Using the ICON model, numerical simulations of different synoptic systems (6 real cases) are conducted for the area covering central Europe. Comparison of the clouds and precipitation properties under different CCN size distributions is done for concluding about aerosol effects on deep convective systems. The simulated results, which are classified to weak and strong synoptic forcing, point on increased total cloud water and decreased total surface rain, with increased CCN concentration and narrower size distribution. The precipitation response is stronger for weakly forced cases. Explanations for these results are suggested by analysis of different hydrometeors types in the simulations and the related formation mechanisms. Less efficient collision-coalescence is demonstrated in more polluted cases (suppressed warm rain formation) and stronger rain evaporation at low levels. The simulated results also show a negative effect of aerosol on the convective intensity meaning there is no convective invigoration. This work examines the interaction of the clouds with their thermodynamic environment as well showing the impact of precipitation on the environmental instability. To summarize, it is a very interesting and valuable work dealing with an important subject which is still not fully understood. We thank the reviewer for this positive comment.

I have a few comments that should be addressed before publication:

1. This study used a bulk microphysical scheme with a saturation adjustment assumption. This method is limited in its ability to simulate rightly the aerosol effect on warm cloud processes. First, the condensation efficiency cannot be accurately described by a saturation adjustment scheme. It was shown that the supersaturation values in clouds depend heavily on the aerosol loading (Pinsky et al., 2013, Seiki and Nakajima, 2014, Dagan et al., 2015). This major effect is neglected in this work. Another major effect is the aerosol impact on the drops' effective terminal velocity (Koren et al., 2015). A bulk scheme is limited in its ability to describe the full range of terminal velocities and so it neglects this major effect too. The authors should regard this major issue and the limitations of the method used here for this type of study should be discussed in more details.

We agree with the reviewer that an explicit formulation of supersaturation might be beneficial. However, this option is not available in the ICON model. As stated by Seifert and Beheng (2006), all clouds, except extremely maritime ones, relax rapidly to the thermodynamic equilibrium between water vapor and water drops. Thus, applying the standard saturation adjustment technique to treat condensational growth seems to be appropriate in almost all cases. Moreover, many other models used for investigating aerosol-cloud interactions also use a saturation adjustment, e.g. COSMO\_ART, ICON\_ART or WRF (e.g. the Morrison scheme). Stensrud (2009) surveys that all single-moment bulk microphysical schemes and most double-moment bulk schemes use bulk condensation, i.e. saturation adjustment instead of saturation prediction. There are a number of studies in the literature where the same microphysics scheme with saturation adjustment has been successfully used for investigating aerosol-cloud interactions, e.g.:

Noppel, H., U. Blahak, A. Seifert, and K. D. Beheng, 2010: Simulations of a hailstorm and the impact of CCN using an advanced two-moment cloud microphysical scheme. Atmos. Res., 96, 286–301.

Seifert, A., C. Köhler, and K. Beheng, 2012: Aerosol-cloud-precipitation effects over Germany as simulated by a convective-scale numerical weather prediction model. Atmos. Chem. Phys., 12, 709–725.

Rieger, D., Steiner, A., Bachmann, V., Gasch, P., Foerstner, J., Deetz, K., Vogel, B., and Vogel,
H.: Impact of the 4 April 2014 Saharan dust outbreak on the photovoltaic power generation in
Germany, Atmos. Chem. Phys., 17, 13391-13415, https://doi.org/10.5194/acp-17-13391-2017,
2017.

Barthlott, C. and Hoose, C.: Aerosol effects on clouds and precipitation over central Europe in different weather regimes, J. Atmos. Sci., 75, 42474264, https://doi.org/10.1175/JAS-D-18-0110.1, 2018.

Keil, C., Baur, F., Bachmann, K., Rasp, S., Schneider, L., and Barthlott, C.: Relative contribution of soil moisture, boundary-layer and microphysical perturbations on convective predictability in different weather regimes, Q. J. R. Meteorol. Soc., 145, 31023115, https://doi.org/10.1002/ qj.3607, 2019.

Costa-Surs, M. et al.: Detection and attribution of aerosolcloud interactions in large-domain large-eddy simulations with the ICOsahedral Non-hydrostatic model, Atmos. Chem. Phys., 20, 56575678, https://doi.org/10.5194/acp-20-5657-2020, 2020.

Stensrud, D. J., 2009: Parameterization schemes: keys to understanding numerical weather prediction models. Cambridge University Press.

However, recent findings by Lebo et al. (2012) and Grabowski and Morrison (2017) suggest that the use of saturation adjustment has indeed implications for cloud development and surface rain amounts. The use of a saturation adjustment in the study of Lebo et al. (2012) enhances condensation and latent heating at lower levels and limits the potential for an CCN increase to increase buoyancy at mid to upper levels which leads to a small weakening of the convective mass flux in polluted compared to clean conditions. Grabowski and Morrison (2017) assumed clean conditions only and showed that the saturation adjustment produced more cloud buoyancy and stronger updrafts. They also state that the impact on surface precipitation is minor and subsequent studies using models with different representations of cloud microphysics and simulating clouds in different environments are needed.

Lebo, Z. J., H. Morrison, and J. H. Seinfeld, 2012: Are simulated aerosol-induced effects on deep convective clouds strongly dependent on saturation adjustment? Atmos. Chem. Phys., 12 (20), 9941–9964, doi:10.5194/acp-12-9941-2012.

Grabowski, W. W., and H. Morrison, 2017: Modeling condensation in deep convection. J. Atmos. Sci., 74 (7), 2247–2267, doi:10.1175/JAS-D-16-0255.1.

We included these comments in section 2.1:

"However, this technique has been shown to affect cloud development and rainfall through enhanced latent heating at lower levels (Lebo et al., 2012; Grabowski and Morrison, 2017) which could reduce the potential for a CCN increase to increase buoyancy at mid to upper levels (Barthlott and Hoose, 2018). According to Grabowski and Morrison (2017), the impact on surface rain amounts was minor only.

• • •

Other recent studies on aerosol-cloud interactions with the ICON model also make use of the saturation adjustment technique (e.g. Seifert et al., 2012, Rieger et al., 2015; Heinze et al., 2017; Costa-Suros et al., 2020, Rybka et al., 2021)."

**Regarding the terminal velocities:**

In the double-moment scheme used in ICON, sedimentation is considered using the corresponding number and mass weighted mean fall velocities. The individual terminal fall velocities are calculated using an empirical relation similar to Rogers et al. (1993), but including an increase of the terminal fall velocity with height (Seifert and Beheng, 2006a). An exponential size distribution for the raindrop ensemble is used and the equation for the individual fall velocity is then integrated to get the weighted fall velocity. We believe that this technique is suitable for our purposes, as the terminal velocity does vary with the size distribution and self-collection or accretion is simulated in a meaningful way.

We added this text in the model description:

"Sedimentation is considered using the corresponding number and mass weighted mean fall velocities with the terminal velocity depending on the mean drop diameter (Seifert and Beheng, 2006a)."

2. This study examines the effects of changes in CCN and CDSD size distributions on deep convective systems. It examines both warm and cold processes. Nevertheless, there is no description or treatment of changes in IN size distribution. If there are changes in aerosol properties it will affect the IN properties and hence the mixed and cold processes. This issue should be explained in the manuscript regarding the treatment of IN in the model and its consequences on the results. We missed to include this information and now added these sentences in the model description section:

"Heterogeneous ice nucleation in the immersion and deposition nucleation modes is calculated based on mineral dust concentrations described in Hande et al. (2015), whereas homogeneous ice nucleation is treated following Kärcher and Lohmann (2002) and Kärcher et al. (2006). The number of ice nucleating particles is not varied in this study, as we solely focus on the impact of different CCN concentrations and CDSD shape parameters."

Thus, the differences in the cold rain processes presented in this study are a result of different warm rain processes, which affect (amongst others) the availability of water vapor, the amount of super-cooled liquid water, and the freezing of cloud and rain droplets.

3. The terminology used in the paper for describing the aerosol loading is very confusing (maritime, continental, polluted). The use of maritime and continental can regard the thermodynamic conditions as well and that why it is confusing. I suggest to change it to clean, intermediate, polluted and highly polluted and to use it in a consistent way throughout the paper.

The terminology originates from the Seifert-Beheng double-moment scheme and has also been used in earlier papers using the same scheme (e.g. Keil et al. 2019, Schneider et al. 2019, Barthlott and Hoose 2018). In order to be consistent with those papers, we like to keep our terminology in that way. However, the reviewer is right about the fact that is not used consistently in the paper and sometimes we write "clean" instead of "maritime". We therefore replaced "clean" with "maritime" throughout the entire manuscript and rephrased the introduction to make it clearer: "In addition, different aerosol amounts ranging from low CCN concentrations (representing maritime conditions) to very high CCN concentrations (representing continental polluted conditions) are assessed."

The wording "clean environment" and "clean air" was not changed.

4. In order to validate the model simulations there is a need to compare it to measurements. I suggest to add a figure which is similar to fig. 4 that will present observed accumulated rain or some other cloud properties for the 6 cases. This will enable estimation of the validity of the simulated results.

A systematic model validation is not within the scope of the present study, because we solely focus on the sensitivity of the model to different CCN concentrations and shape parameters. Although the use of double-moment scheme has been shown to improve quantitative precipitation forecasting in some studies, the German Weather Service still uses a single-moment scheme due to computational costs. However, we agree with the reviewer that our simulations must reproduce the observed weather characteristics at least in a qualitative way. Therefore we compared the simulated 24-h precipitation of our reference runs to observations from a radar network combined with surface stations (RADOLAN, Radar Online Adjustment). It combines weather radar data with hourly surface precipitation observations of about 1300 automated rain gauges to get quality-controlled, high-resolution (1 km) quantitative precipitation estimations. The simulated precipitation of the reference runs generally show good agreement with observations (Fig. R.1) and the weather patterns of the days analysed are captured reasonably well. Even if not all precipitation is simulated at the right place with the correct intensity, we believe that these runs serve as a good basis for our analysis, as the focus of this study is the model sensitivity and not a quantitative evaluation or model improvement.

We added this text at the end of section 2.2:

"The intercomparison of the simulated precipitation amounts to Radar-derived precipitation (not shown) reveals that although the exact location of individual convective cells are not always simulated, there is an overall good agreement between observations and simulations. As the model succeeds reasonably well in reproducing the observed weather characteristics, we conclude that these reference runs serve as a good basis for our sensitivity studies."

5. Fig. 6: The meaning of the shading is not explained in the figure caption so the figure is unclear. It should be added.

Thank you for pointing that out, we added the required information in the caption text.

6. The idea of considering the interaction of the convective clouds and the available instability as a type of "lifetime effect" is problematic as it treats a whole cloud system and not a single cloud (as the lifetime effect). This idea should be examined again and any way it should be explained better.

The cloud lifetime effect was initially formulated for one particular cloud regime and the graphical illustration of this effect (e.g. Fig. 1 in Stevens and Feingold (2009)) indicates a single cloud as a representation of the average response of a field of clouds. We therefore believe that the lifetime effect is not restricted to one single cloud. However, we agree with the reviewer that our idea might be misunderstood by the reader and therefore removed the respective sentences in section 3.2 and the Summary. We replaced it with this formulation in Section 3.1:

"As this interaction has an impact on the lifecycle of the convective clouds through decay and intensification, the entire lifetime of the cloud field is also affected, which highlights the complex interactions between thermodynamic and microphysical processes."

Stevens, B., Feingold, G. Untangling aerosol effects on clouds and precipitation in a buffered

---

## Author Response (AR2)

**Responses to the reviewers**

Importance of aerosols and shape of the cloud droplet size distribution for convective clouds and precipitation

by C. Barthlott, A. Zarboo, T. Matsunobu, and C. Keil                    December 7, 2021

We thank both reviewers for reading the revised manuscript again. We have carefully considered all remaining comments of Reviewer 1 and changed the manuscript accordingly. Please find below our responses in blue.

**Reviewer 1**

The paper is certainly improved over the original submission, but I do have several additional comments.

Main comment:

1. Lines 237-240: But the sensitivity is not small for all strong forcing cases, only the one for which the heaviest precipitation occurs early. Perhaps a better way to ask my original question is, would you see the same negligible sensitivity in the red strong forcing case and any of the weak forcing cases if the red forcing case were started at say 18Z and if the weak forcing cases were started at say 10Z? Or, would you see increased sensitivity for the yellow strong forcing case if that simulations were started 12 hours earlier? In other words, is the sensitivity mostly a result of natural model spread that results from small changes in the pre-storm environment, or is it really that the CCN/shape parameter are having their biggest impacts during the precipitation events? If the latter, then I'd imagine that the sensitivity would not be especially sensitive to the model start time.

   We did not mention that the sensitivity in all strong forcing cases is small, we specifically mentioned the case of 2 June 2016 only:

   *"The comparably small spread in precipitation intensities for both the CCN and shape parameter runs during the nighttime precipitation maximum on 2 June 2016 could be explained by the fact that particularly in cases with strong synoptic forcing, clouds act more like a buffered system and the response of precipitation to these microphysical uncertainties remains small."*

   Two of the three strong forcing cases reveal a weaker sensitivity throughout the entire simulation time (13 September 2013 and 2 June 2016), such a behaviour was also found in previous work studying aerosol–cloud interactions with the COSMO model. The case of 11 June 2019, however, shows a strong sensitivity during the time of the precipitation maximum in the afternoon/evening. The stronger sensitivity cannot be attributed to the fact that precipitation intensities are high, because the case of 2 June 2016 reveals similar high rain intensities in the early morning hours. However, the sensitivity to CCNs or the shape parameter is small in that time frame. We therefore followed the reviewer's idea to relate the sensitivity to the model initialization time. However, we must state that it is not ideal to start the model at the time of a precipitation event or shortly before due to spin-up effects. We conducted all 8 model runs for 11 June 2019, but with an initial time of 18 UTC. The resulting sensitivity of the precipitation rates is given in Fig. R.1. It can be seen that the sensitivity is smaller as in the runs initialized at 00 UTC. We therefore believe that major precipitation events simulated in the first hours of integration are less affected by our microphysical perturbations as do runs with longer lead times.

[Figure]

Figure R.1: Domain-averaged precipitation rates for the strong forcing of 11 June 2019 for a model initialization at 1800 UTC. The color-coded areas indicate the range between the minimum and maximum precipitation rate for all CCN sensitivities in (c) and all shape parameter sensitivities in (d). The black lines indicate the respective reference run with continental CCN concentration and a shape parameter of 0.

We modified the text in the manuscript as follows:

*"The comparably small spread in precipitation intensities for both the CCN and shape parameter runs during the nighttime precipitation maximum on 2 June 2016 could be explained by the fact that  this maximum occurs during the first hours of simulation. In that time, spin-up effects and the adjustment to the driving coarser-scale model are still in effect, which could dampen the impacts of the microphysical uncertainties assessed here. A similar smaller impact of microphysics perturbations at short lead times was found in further sensitivity experiments for 11 June 2019 initializing the model at 18:00 UTC (not shown)."*

We therefore hypothesize that although two out of three strong forcing cases generally reveal a weaker sensitivity, the short lead time for the nighttime precipitation maximum could buffer potential aerosol or shape parameter impacts for that case. We hope that this is sufficient and leave a more systematic evaluation of this effect to future work.

Minor comments:

2. Lines 64-80: Thanks to the authors for including this discussion. The authors may want to combine equations (3) and (4) since (4) follows from (3). As written, it is not obvious that this is the case, rather, the equations seem to be mysteriously equivalent. A reader familiar with distributions will know that $N0$ is equal to $A$ in line 77, and that $a = 0.124$ m kg$^{-b}$ arises from $(\pi/6\rho)^{-1/3}$, which is the factor appearing in multiple terms in line 77. But as is, the relationship between (3) and (4) is rather obscured by the use of different notations. I'd recommend making this discussion more transparent.

We included the equations (2)–(4) in the revised manuscript as the reviewer requested a distinction between size distributions as a function of particle mass $x$ and diameter $D$. Especially the approach in Eq. (2) may not be known to a reader unfamiliar with size distributions. But as Eqs. (3) and (4) seem to be confusing, we have again deleted Eqs. (2) and (3). Now, we only refer to the equations given in Khain et al. (2015) and hope that the use of one notation is

sufficient.

3. I want to make the point again that if a typical range of f(D) shape parameters is 0-14 (Line 84), that this corresponds to a typical range of $f(x)$ shape parameters of -2 to 4 (using the conversion equation in line 77). The authors largest choice of $f(x)$ shape parameter, 8, corresponds to a f(D) shape parameter of 26, which is arguably unrealistic. I don't think anything major in the paper needs to be changed, but I think the authors should discuss the realism of their range. It may be helpful to explicitly include the converted range that I've mentioned here.

We agree with the reviewer that our shape parameter values based on $f(x)$ should be converted for $f(D)$ for better comparison to literature values based on $f(D)$ as well. Therefore, we expanded Tab. 1 with $\nu'$. A shape parameter value of $\nu' = 26$ might not be realistic and too large compared to observational values (0–14). The reason for letting $\nu'$ have such a wide range is to show how large the response of simulated clouds and precipitation could be in such extreme conditions. Furthermore, there is another cloud particle class often used in ICON with a shape parameter of 1 and a dispersion parameter of 1. The resulting CDSD looks broadly similar to the one with $\nu = 8$ and $\mu = 1/3$ (Fig. R.2), differences are mainly obvious for small droplet sizes.

[Figure]

Figure R.2: Example cloud droplet size distributions.

4. Lines 163-166: Thanks also for this list of processes that are impacted by the shape parameter. However, previously the authors stated that they are using a saturation adjustment scheme, so the cloud droplet shape parameter and DSD properties can't possibly have an impact on condensation rates (and also evaporation rates?) in the model. Please can the authors review this list carefully?

Thanks for pointing that out, the reviewer is right about the fact that the condensation rates are not directly influenced by the shape of the cloud droplet size distribution. In the course of the model run, however, we do see differences between condensation rates resulting from microphysical impacts on the thermodynamic environment. Evaporation is definitely influenced by the shape of the size distribution. We therefore deleted condensation from our list of processes.

5. I believe the microphysics scheme being used in this study is mostly the same as that described by Seifert and Beheng (2006), but I can't find that explicitly mentioned anywhere in the paper. Can the authors please provide a general reference for the scheme, if appropriate?

Yes, we use the double-moment scheme of Seifert and Beheng. Maybe the reviewer missed it, it was already mentioned in the model description part (section 2.1 on page 5):

*"For the simulation of aerosol effects on mixed-phase clouds, we use the double-moment microphysics scheme of Seifert and Beheng (2006a) which enables the use of four different CCN concentration assumptions."*

6. Eq (5): I should have asked this the first time. Can the authors provide the expression used to calculate the effective radius in terms of the distribution parameters introduced in the introduction? I think this would help readers to explicitly see how effective radius is modulated by the shape parameter.

In section 3.6, we mention the equation we use to compute the effective radius:

$$r_{\text{eff}} = \frac{\int r^3 n(r)\mathrm{d}r}{\int r^2 n(r)\mathrm{d}r},$$

where $r$ is the droplet radius and $n(r)$ the cloud droplet size distribution. As we integrate the size distribution over the droplet radius, we believe that the influence of the shape parameter is understandable to the reader. The method to actually calculate the size distribution is given in Seifert and Beheng (2006a):

$$f(x) = Ax^\nu \exp(-\lambda x^\mu)$$

The coefficients $A$ and $\lambda$ can be expressed by the number ($N$) and mass ($L$) densities:

$$A = \frac{\mu N}{\Gamma(\frac{\nu+1}{\mu})}\lambda^{\frac{\nu+1}{\mu}} \text{ and } \lambda = \left[\frac{\Gamma(\frac{\nu+1}{\mu})}{\Gamma(\frac{\nu+2}{\mu})}\overline{x}\right]^{-\mu}$$

$$\overline{x} = \frac{L}{N} \text{ mean particle mass}$$

$$f(x) = \frac{N}{\overline{x}}\left[\frac{x}{\overline{x}}\right]^\nu \frac{\mu}{\Gamma\left(\frac{\nu+1}{\mu}\right)}\left[\frac{\Gamma\left(\frac{\nu+2}{\mu}\right)}{\Gamma\left(\frac{\nu+1}{\mu}\right)}\right]^{\nu+1} \times \exp\left\{-\left[\frac{\Gamma\left(\frac{\nu+2}{\mu}\right)}{\Gamma\left(\frac{\nu+1}{\mu}\right)}\frac{x}{\overline{x}}\right]^\mu\right\}$$

Using a power law for the diameter-mass relation $D(x) = ax^b$ ($a = 0.124$ m kg$^{-b}$, $b = 1/3$), we can transform the equation from mass $x$ to particle diameter $D$ (or radius $r$):

$$f(D) = f(x)/(dD/dx) \quad D = ax^b \quad dD/dx = bax^{(b-1)}$$
$$= f(x)/bax^{(b-1)}$$

We included a reference to the equations in the Seifert and Beheng paper and hope that this additional information is sufficient.

**Reviewer 2**

I have read the reply to my review and I think the authors did a very good job with answering my comments. I agree with their answers and I think the manuscript can be published in its current form.

We thank the reviewer for evaluating our reply to his comments.

**Additional corrections:**

- We changed the cloud albedo $A$ to $A_c$ because $A$ was already used in Eq. 1 for the cloud droplet size distribution.

---

## Author Response (AR3)

**Responses to the reviewers**

Importance of aerosols and shape of the cloud droplet size distribution for convective clouds and precipitation

by C. Barthlott, A. Zarboo, T. Matsunobu, and C. Keil                    January 24, 2022

We thank the reviewer for reading the revised manuscript again.

**Reviewer 1**

Thank you to the authors for addressing my previous comments. My only concern still is the list of processes impacted by the shape parameter. Based on the response, I'm still worried that the list contains too many processes. Only those processes that are directly impacted through use of the shape parameter in the calculation of the instantaneous process rate should be included. The authors state that evaporation is definitely impacted, but the way the response was worded, I can't tell if it is a direct impact or an indirect impact. I can't find any documentation of cloud droplet evaporation in this scheme and so can't verify if it is a direct impact. All I ask is that the authors check the code for all processes listed to make sure that the droplet shape parameter is actually used and make any changes if necessary. Otherwise the paper can be published in its present form.

We have rephrased the list of processes which are influenced by the shape parameter as suggested, the sentence now reads:
*"The size distribution of the cloud droplets has a substantial impact on the simulation results, as various microphysical processes depend either directly on the shape parameter (e.g. autoconversion, self collection) or indirectly (e.g. accretion, sedimentation, evaporation, riming, melting)."*